# INT: An Inequality Benchmark for Evaluating Generalization in Theorem Proving

**Yuhuai Wu**[*], **Albert Qiaochu Jiang**[*], **Jimmy Ba & Roger Grosse**
University of Toronto & Vector Institute
`{ywu, ajiang, jba, rgrosse}@cs.toronto.edu`

## Abstract

In learning-assisted theorem proving, one of the most critical challenges is to generalize to theorems unlike those seen at training time. In this paper, we introduce INT, an INequality Theorem proving benchmark designed to test agents' generalization ability. INT is based on a theorem generator, which provides theoretically infinite data and allows us to measure 6 different types of generalization, each reflecting a distinct challenge, characteristic of automated theorem proving. In addition, INT provides a fast theorem proving environment with sequence-based and graph-based interfaces, conducive to performing learning-based research. We introduce baselines with architectures including transformers and graph neural networks (GNNs) for INT. Using INT, we find that transformer-based agents achieve stronger test performance for most of the generalization tasks, despite having much larger out-of-distribution generalization gaps than GNNs. We further find that the addition of Monte Carlo Tree Search (MCTS) at test time helps to prove new theorems.

## 1 Introduction

Advances in theorem proving can catalyze developments in fields including formal mathematics (McCune, 1997), software verification (Darvas et al., 2005), and hardware design (Kern and Greenstreet, 1999). Following its recent success across other application domains, machine learning has significantly improved the performance of theorem provers (Bansal et al., 2019; Bridge et al., 2014; Gauthier et al., 2018; Huang et al., 2019; Irving et al., 2016; Kaliszyk et al., 2018; Lee et al., 2020; Loos et al., 2017; Urban et al., 2011; Wang and Deng, 2020; Yang and Deng, 2019; Li et al., 2020; Rabe et al., 2020; Polu and Sutskever, 2020). Two key factors that make theorem proving particularly challenging for ML are data sparsity and that it requires out-of-distribution generalization.

Firstly, due to the difficulty of formalizing mathematics for humans, manually generated formal proofs are necessarily expensive. Typical formal mathematics datasets contain thousands (Huang et al., 2019) to tens-of-thousands (Yang and Deng, 2019) of theorems — orders of magnitude smaller than datasets that enabled breakthroughs in areas such as vision (Deng et al., 2009) and natural language processing (Rajpurkar et al., 2016). Secondly, the assumption frequently made in machine learning that each data point is identically and independently distributed does not hold in general for theorem proving: interesting problems we want to prove are non-trivially different from those we have proofs for. Hence, the out-of-distribution generalization ability is crucial.

Synthetic datasets that rely on procedural generation provide a potentially unlimited amount of data. Well-designed synthetic datasets have been shown to help understand the capabilities of machine learning models (Johnson et al., 2017; Ros et al., 2016; Weston et al., 2016). With the goal of alleviating the data scarcity problem and understanding out-of-distribution generalization for theorem proving, we introduce INT. INT is a synthetic INequality Theorem proving benchmark designed for evaluating generalization. It can generate a theoretically unlimited number of theorems and proofs in the domain of algebraic equalities and inequalities. INT allows tweaking of its problem distribution along 6 dimensions, enabling us to probe multiple aspects of out-of-distribution generalization. It is accompanied by a fast proof assistant with sequence and graph-based interfaces. A common reservation to hold for synthetic datasets is one of realism: can synthetic data help to prove realistic

---

[*]: equal contribution

theorems? Polu and Sutskever (2020) adopted our generation method and showed that augmentation of 1% of synthetic theorems in training helped to complete 2.3% more proofs on Metamath (Megill and Wheeler, 2019). This demonstrates the usefulness of INT in real mathematics.

Time and memory requirements for the proof assistant have often been an obstacle for using theorem provers as RL environments. Most existing proof assistants require a large software library to define numerous mathematical theorems, leading to slow simulation. Therefore, a key design objective for INT was to be lightweight and swift. Taking advantage of the limited scope of inequality theorems, we load a minimal library and achieve fast simulation. Reducing the simulation overhead allows for experimentation with planning methods such as MCTS which requires many calls to a simulator.

We summarize the contributions of this paper as follows:

1. We make, to the best of our knowledge, the first attempt to investigate an important question in learning-assisted theorem proving research, i.e., can theorem provers generalize to different problem distributions? We introduce INT for evaluating six dimensions of generalization.

2. We introduce and benchmark baseline agents for the six types of generalization tasks in INT. We find that transformer-based agents' generalization abilities are superior when training and test data are drawn from the same distribution and inferior in out-of-distribution tasks in INT, compared to GNN-based agents. Surprisingly, despite larger generalization gaps, transformer-based agents have favorable test success rates over GNN-based ones in most cases.

3. We find that searching with MCTS at test time greatly improves generalization.

## 2 RELATED WORKS

**Automatic and Interactive Theorem Proving.**   Modern Automatic Theorem Provers (ATPs) such as E (Schulz, 2013) and Vampire (Kovács and Voronkov, 2013) represent mathematical theorems in first-order logic and prove them with resolution-based proof calculi. On the other hand, Interactive Theorem Provers (ITPs) allow human formalization of proofs. This perhaps makes them more suitable for biologically inspired methods such as machine learning. Famous ITPs include Isabelle (Paulson, 1986), Coq (Barras et al., 1999), LEAN (de Moura et al., 2015), and HOL Light (Harrison, 1996).

**Learning-assisted Theorem Proving.**   Theorem provers have been improved by supervised learning (Urban et al., 2011; Bridge et al., 2014; Irving et al., 2016; Loos et al., 2017; Wang et al., 2017; Rocktäschel and Riedel, 2017; Bansal et al., 2019; Gauthier et al., 2018; Huang et al., 2019; Yang and Deng, 2019; Kaliszyk and Urban, 2015; Polu and Sutskever, 2020; Li et al., 2020; Rabe et al., 2020; Jakubuv and Urban, 2019; Olsák et al., 2020; Jakubuv et al., 2020; Kaliszyk et al., 2015; Gauthier and Kaliszyk, 2015). Wang et al. (2017) used graph embeddings to represent logic formulas and achieved state-of-the-art classification accuracy on the HolStep dataset (Kaliszyk et al., 2017). Reinforcement learning (RL) was employed in (Zombori et al., 2019; Gauthier, 2019; 2020). Kaliszyk et al. (2018) combined MCTS with RL to prove theorems with connection tableau. Notably, GPT-f (Polu and Sutskever, 2020) adopts our INT generation method for dataset augmentation.

**Datasets for Theorem Proving.**   There have been many formal mathematical libraries (Megill and Wheeler, 2019; Rudnicki, 1992; Gauthier, 2019). Formalized mathematical theorems include the Feit-Thompson theorem (Gonthier et al., 2013) and the Kepler Conjecture (Hales et al., 2017). The largest human formal reasoning dataset is IsarStep (Li et al., 2020), where they mined the archive of formal proofs and brought together 143K theorems in total. These works rely on human efforts to formalize theorems, which leads to small to moderate-sized datasets. There have been studies on synthesizing theorems (Urban, 2007; Urban et al., 2008; Piotrowski and Urban, 2018; Gauthier et al., 2017; 2016; Chvalovský et al., 2019; Lenat, 1976; Fajtlowicz, 1988; Colton, 2012; Johansson et al., 2014) It is worth mentioning that there have been a few approaches (Urban and Jakubv, 2020; Wang and Deng, 2020) on neural theorem synthesizers. Our theorem generator INT is designed to be capable of creating an infinite number of theorems, as well as benchmarking the generalization ability of learning-assisted theorem provers.

## 3 THE INT BENCHMARK DATASET AND PROOF ASSISTANT

Our INT benchmark dataset provides mathematical theorems and a means to study the generalization capability of theorem provers. For this purpose, we need control over the distribution of theorems:

this is achieved by a highly customizable synthetic theorem generator. We used a set of ordered field axioms (Dummit and Foote, 2004) to generate inequality theorems and a subset of it to generate equality theorems. Details of the axiomization schemes can be found in Appendix A. The code for generating theorems and conducting experiments is available at `https://github.com/albertqjiang/INT`.

## 3.1 TERMINOLOGY

The axiom combination of a proof refers to the set of axioms used in constructing it. The sequence of axioms applied in order in the proof is called the axiom order. For example, let $A, B, C$ denote three unique axioms, and their order of application in a proof be $[B, B, A, C]$. In this case, the axiom combination is the set $\{A, B, C\}$ and the axiom order is the sequence $[B, B, A, C]$. An initial condition is a (usually trivial) logic statement (e.g. $a = a$) to initiate the theorem generation process. The degree of an expression is the number of arithmetic operators used to construct it. For example, degree$(a) = 0$ while degree$(((a * c) * b)^2) = 3$.

## 3.2 INT ASSISTANT

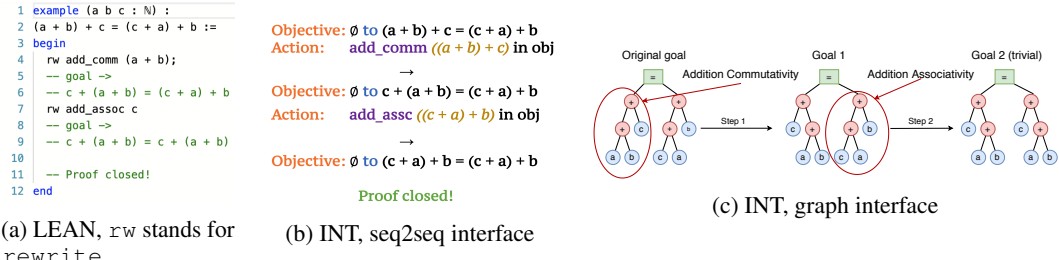

(a) LEAN, `rw` stands for `rewrite`

(b) INT, seq2seq interface

(c) INT, graph interface

Figure 1: A proof of $a + b + c = c + a + b$ in LEAN and INT, with seq2seq and graph interfaces.

We built a lightweight proof assistant to interact with theorem provers. It has two interfaces, providing theorem provers with sequential and graph representations of the proof state, respectively.

A problem in INT is represented by a goal and a set of premises (e.g. $a + 0 = a, \varnothing$), which are mathematical propositions. The INT assistant maintains a proof state composed of the goal and the proven facts. The proof state is initialized to be just the goal and premises of the theorem. A proof is a sequence of axiom-arguments tuples (e.g. $[(\text{AdditionZero}, [a + 0])]$). At each step of the proof, a tuple is used to produce a logical relation in the form of assumptions $\rightarrow$ conclusions (e.g. $\varnothing \rightarrow a + 0 = a$). Then, if the assumptions are in the proven facts, the conclusions are added to the proven facts; if the conclusions include the goal, the unproven assumptions will become the new goal. The assistant considers the theorem proven, if after all steps in the proof are applied, the goal is empty or trivial.

In Figure 1, we present the same proof in LEAN (de Moura et al., 2015) and INT assistants. They both process proofs by simplifying the goal until it is trivial. The INT assistant's seq2seq interface (Figure 1b) is very similar to that of LEAN (Figure 1a) with the `rewrite` tactic. An action is composed of an axiom followed by argument names and their positions in the proof state. `in obj` indicates that the arguments can be found in the objective. The graph interface (Figure 1c) of the INT assistant allows theorem provers to chose axiom arguments from the computation graphs of the proof state by node. We can view theorem proving with this interface as a graph manipulation task.

INT assistant provides fast simulation. To demonstrate this, we produced 10,000 typical proof steps in both interfaces, 40-character-long on average. We executed them with HOL Light (Harrison, 1996) and INT assistant. The average time it takes per step is 7.96ms in HOL Light and 1.28ms in INT, resulting in a 6.2× speedup. The correctness of the proofs is ensured by a trusted core of fewer than 200 lines of code.

### 3.3 THEOREM GENERATOR

One of the main contributions of this paper is to provide a generation algorithm that is able to produce a distribution of non-trivial synthetic theorems given an axiom order. Generating theorems by randomly sampling axiom and argument applications will often yield theorems with short proofs. Instead, we write production rules for axioms in the form of transformation and extension rules. With these production rules, we can find arguments and new premises required for longer proofs.

We provide the theorem generation algorithm in Algorithm 1. The general idea of the algorithm is to morph a trivial logic statement into one that requires a non-trivial proof; we call this statement the core logic statement. We initiate the core logic statement $C_0$ to be one of the initial conditions. At step $t$ of the generation process, we are given an axiom $a_t$ specified by the axiom order. We apply the MORPH function associated with the axiom $a_t$ to $C_{t-1}$ and derive a new logic statement $C_t$ and corresponding premises $P_t$. The key design idea in the MORPH function is to ensure that the newly

---

**Algorithm 1** Theorem Generator

1: **function** GENERATE_THEOREM(initial conditions $\mathcal{I}$, axiom order $A$)
2:     Axiom order length $L = \text{len}(A)$.
3:     Initialize core logic statement $C_0 \sim Uniform(\mathcal{I})$, and the set of premises $P = \{C_0\}$.
4:     **for** $t \leftarrow 1$ to $L$ **do**
5:         Get axiom $a_t \leftarrow A[t]$.
6:         Get new logic statement and premises: $C_t, P_t \leftarrow$ MORPH $(a_t, C_{t-1})$.
7:         Add new premises to the set of all premises: $P \leftarrow P \cup P_t$.
8:     **end for**
9:     **return** $C_L, P$
10: **end function**

---

generated logic statement and the premises form the implication $C_{t-1}, a_t, P_t \rightarrow C_t$ (see Appendix B for details). Therefore, we can chain the implications from all steps together to obtain a proof whose length is the axiom order: $C_0, \{a_t, P_t\}_{t=1}^{L} \rightarrow C_L$, where $L$ denotes the length. The last core logic statement $C_L$ and its premises $C_0, \{P_t\}_{t=1}^{L}$ are returned as the theorem generated. Below we show a step-by-step example of how a theorem is generated with our algorithm.

---

**A worked example**

Use Algorithm 1 to generate a theorem with initial conditions $\mathcal{I}$: $\{a = a, b = b, c = c, d = d, e = e\}$ and axiom order $A$: [AdditionAssociativity (AA), AdditionCommutativity (AC), EquivalenceImplies-DoubleInequality (EIDI), FirstPrincipleOfInequality (FPI)].

Core logic statement $C_0 \sim Uniform(\mathcal{I})$ : $a = a$.
Step 1: $a_1 =$ AA. $C_1$: $a + (b + c) = (a + b) + c$, $P_1 = \varnothing$.
Step 2: $a_2 =$ AC. $C_2$: $a + (b + c) = (b + a) + c$, $P_2 = \varnothing$.
Step 3: $a_3 =$ EIDI. $C_3$: $a + (b + c) \geq (b + a) + c$, $P_3 = \varnothing$.
Step 4: $a_4 =$ FPI. $C_4$: $(a + (b + c)) + d \geq ((b + a) + c) + e$, $P_4 = \{d \geq e\}$.

Theorem generated: Given $d \geq e$, prove $a + (b + c) + d \geq b + a + c + e$.

---

With recorded axiom and argument applications, we can synthesize proofs to the theorems. The proofs can be used for behavior cloning. Appendix E shows statistics of the generated proofs, including the distribution of length of theorems in characters, the distribution of axioms, and the distribution of the number of nodes in proof state graphs.

## 4 EXPERIMENTS

Our experiments are intended to answer the following questions:

1. Can neural agents generalize to theorems: 1) sampled from the same distribution as training data, 2) with different initial conditions, 3) with unseen axiom orders, 4) with unseen axiom combinations, 5) with different numbers of unique axioms, 6) with shorter or longer proofs?

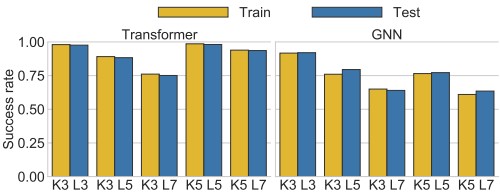 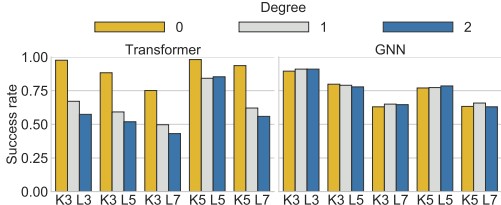

Figure 2: Proof success rates on problems generated with different $K$ and $L$ parameters. **Left:** When the IID assumption holds, the success rate decreases as the two generation parameters $K$ and $L$ are increased. **Right:** All agents are trained on degree-0 problems and evaluated against problems of degree 0, 1, and 2. We find that transformer-based agents deteriorate in performance as the test problems become more complex than training problems. For GNN-based agents, there are no obvious trends as to how the proof success rate changes as the degree of the initial entities is varied.

2. How do different architectures (transformer vs. GNN) affect theorem provers' in-distribution and out-of-distribution generalization?

3. Can search at test time help generalization?

## 4.1 EXPERIMENT DETAILS

In the following experiments, we used the proofs generated by the INT generator to perform behavior cloning. We then evaluated the success rates of trained agents in a theorem proving environment. We denote the cardinality of an axiom combination as $K$ and the length of a proof as $L$. In the worked example, $K = 4$ and $L = 4$. For each theorem distribution, we first generated a fixed test set of 1000 problems, and then produced training problems in an online fashion, while making sure the training problems were different from the test ones. For each experiment, we generated 1000 problems and performed 10 epochs of training before generating the next 1000. We ran 1500 such iterations in total, with 1.5 million problems generated. We used the Adam optimizer (Kingma and Ba, 2015). We searched over the learning rates $\{10^{-5}, 3 \cdot 10^{-5}, 10^{-4}, 3 \cdot 10^{-4}\}$ in preliminary experiments and found $10^{-4}$ to be the best choice, which was used for following experiments. We used one Nvidia P100 or Tesla T4 GPU with 4 CPU cores for training. For each experiment, we ran 2 random seeds, and picked the one with higher validation success rates for test evaluation. Since this paper focuses on inequalities, all figures and tables in the main text are based on results from the ordered-field axiomization. We also include results of GNN-based agents on equalities in Appendix G.

## 4.2 NETWORK ARCHITECTURES

In this section, we introduce four baselines built on commonly used architectures: Transformers (Vaswani et al., 2017), Graph Neural Networks (GNNs), TreeLSTMs (Tai et al., 2015) and Bag-of-Words (BoWs). In preliminary experiments, we found Graph Isomorphism Networks (GINs) (Xu et al., 2019) to have performed the best among several representative GNN architectures. So we used GIN as our GNN of choice. Transformers interact with the INT proof assistant through the seq2seq interface while the other baselines through the graph interface.

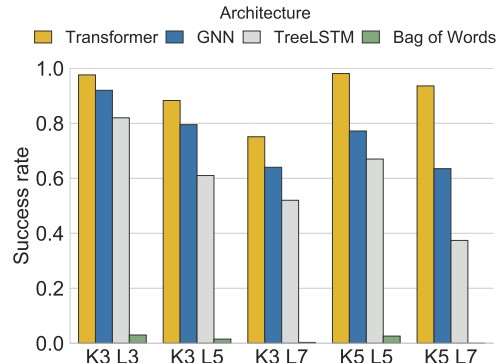

Figure 3: Proof success rates on test problems generated with $K$ and $L$ settings. Transformer and GNN perform well; TreeLSTM has mediocre performance; and Bag-of-Words performs poorly: it cannot prove more than 5% of problems.

For sequence-to-sequence training, we used a character-level transformer architecture with 6 encoding layers and 6 decoding layers. We used 512 embedding dimensions, 8 attention heads and 2048 hidden dimensions for position-wise feedforward layers. We used dropout with rate 0.1, label smoothing with coefficient 0.1, and a maximum 2048 tokens per batch. The library fairseq (Ott et al., 2019) was used for its implementation.

Table 1: **Left:** Average success rates (in %) of agents trained on different numbers of axiom orders. **Right:** Average success rates (in %) of agents trained on different numbers of axiom combinations.

| # Axiom orders | 100 | | 500 | | 2000 | | 5000 | |
|---|---|---|---|---|---|---|---|---|
| | Train | Test | Train | Test | Train | Test | Train | Test |
| Transformer | 93.2 | 10.0 | 93.4 | **62.8** | 93.6 | **87.9** | 93.7 | **91.8** |
| GNN | 87.6 | **21.1** | 86.6 | 53.6 | 79.0 | 70.4 | 75.7 | 74.7 |

| # Axiom combinations | 25 | | 100 | | 200 | | 300 | |
|---|---|---|---|---|---|---|---|---|
| | Train | Test | Train | Test | Train | Test | Train | Test |
| Transformer | 96.1 | 29.3 | 96.0 | **71.8** | 95.4 | **88.4** | 94.4 | **91.3** |
| GNN | 79.1 | **47.5** | 76.6 | 68.0 | 72.6 | 72.4 | 72.8 | 71.9 |

For data in the graph form, each node in computation graphs corresponds to a character in the formula. We first used a learnable word embedding of dimension 512 to represent each node. We then used 6 GIN layers to encode graph inputs into vector representations, each with 512 hidden dimensions. The graph representation was obtained by taking the sum of all the node embeddings. For the TreeLSTM and the BoW baselines, we used a bidirectional TreeLSTM with 512 hidden dimensions and a BoW architecture to compute the graph representation vectors from node embeddings. The hyper-parameters used were found to be optimal in preliminary experiments. We then proposed axioms conditioned on the graph representations, with a two-layer MLP of hidden dimension 256. Conditioning on the graph representation and axiom prediction, the arguments are selected in an autoregressive fashion. Namely, the prediction of the next node is conditioned on the previous ones. For each argument prediction, we used a one-layer MLP with a hidden size of 256. We used graph neural network libraries Pytorch Geometric (Fey and Lenssen, 2019) for the GIN implementation, and DGL (Wang et al., 2019) for the TreeLSTM implementation.

We trained agents based on architectures mentioned above by behavior cloning on theorems of various length ($L$) and number of axioms ($K$). The success rates for proving 1000 test theorems are plotted in Figure 3. As the BoW architecture did not utilize the structure of the state, it failed miserably at proving theorems, indicating the significance of the structural information. TreeLSTM performed worse than the graph neural network baseline. The transformer and the GNN baselines perform the best among the architectures chosen and they take inputs in sequential and graph forms, respectively. Thus, we used these two architectures in the following experiments to investigate generalization.

### 4.3 Benchmarking Six Dimensions of Generalization

**IID Generalization** In this experiment, the training and test data are independently and identically distributed (IID). The performances of our transformer-based and GNN-based agents are displayed on the left in Figure 2. As can be seen, the performance of agents examined on train and test problems are very similar. The largest difference between train and test success rates is $2\%$ ($K3L7$). Notably, transformer-based agents complete $15.3\%$ more test proofs than GNN-based agents on average.

**Initial Condition** Consider two theorems: (1) $(a + b)^2 = a^2 + b^2 + 2ab$ and (2) $(a + (b + c))^2 = a^2 + (b + c)^2 + 2a(b + c)$. The two problems take the same axioms and the same number of steps to prove. However, the axiom argument complexities are different, which can be seen as a result of varying initial conditions. Can agents trained on problems like (1) prove theorems like (2)?

For an initial condition of the form $X = X$, we use the degree of the entity $X$ to determine the complexity. In this experiment, we trained agents on problems with initial conditions made up of entities of degree 0, and evaluated them on ones of degrees 1 and 2. The results are presented in Figure 2 (b) with various $K$ and $L$. For transformer-based agents, the success rate drops $25.6\%$ on degree-1 problems and $31.5\%$ on degree-2 problems on average. However, for GNN-based agents, the largest generalization gap between training and test success rates is $3\%$ ($K3L5$). This shows that GNN agents can generalize to problems of higher complexities while transformer agents struggle.

**Axiom Orders** Let $A$ and $B$ represent two different axioms. There are multiple orders in which they can be applied in a $K2L3$ problem. $O_1 = [A, A, B]$ and $O_2 = [B, A, B]$ are two examples. Can an agent trained on problems generated with $O_1$ prove theorems generated with $O_2$?

For both architectures, we investigated how well agents can generalize to problems with different axiom orders than those in training. We generated 100, 500, 2000, and 5000 axiom orders to use in the training set for different $K$ and $L$ settings. We evaluated the test success rates on 1000 unseen axiom orders with the corresponding $K$ and $L$ settings and averaged them. The results averaged over different $K$ and $L$ settings are shown on the left of Table 1 (See Appendix G.5 for the full results).

It can be observed in the table that the test success rates rise when we increase the number of axiom orders in the training set. We notice that transformer-based agents have worse generalization than

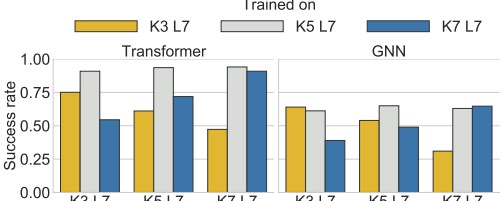 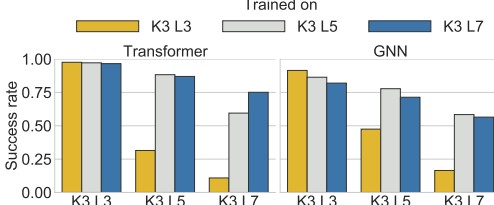

Figure 4: Proof success rates on problems generated with different parameters. **Left:** We keep $L$ the same and vary $K$. The success rate is likely to decrease when the test problems have different $K$ from the training problems. **Right:** We keep $K$ the same and vary $L$. For all agents, the proof success rate is lower on theorems that require longer proofs.

GNN-based ones, as their average generalization gap is larger. This is particularly true when the number of axiom orders in the training set is 100: transformer-based agents can prove only $10.0\%$ of test theorems. Remarkably, they still manage to complete more proofs than GNNs when the number of axiom orders in the training set exceeds 500.

**Axiom Combinations**    Consider three problems provable in the ordered field axiomization (Appendix A): (1) $a^2 \geq 0$, (2) $a * (b + c) = b * a + a * c$, and (3) $a^2 + b^2 - 2ab \geq 0$. Solving (1) requires axiom SquareGEQZero (SGEQZ). Solving (2) requires axiom AdditionMultiplicationDistribution (AMD) and axiom MultiplicationCommutativity (MC). Solving (3) requires axiom SGEQZ and axiom AMD. Notice that all axioms used to prove (3) appear in the proofs of (1) and (2). We ask: can an agent trained on theorems like (1) and (2) prove theorems like (3)?

In this set of experiments, we investigated how well theorem provers can generalize to problems with different axiom combinations than those in training for both architectures. We used 25, 100, 200, and 300 axiom combinations to generate the training set with various $K$ and $L$ settings, and evaluated the agents on test sets generated with 300 unseen combinations. The results averaged over different $K$ and $L$ settings are displayed on the right in Table 1 (See Appendix G.5 for full results). As the number of axiom combinations in training set increases, the generalization gap decreases and test success rate improves. The transformer-based agents have larger generalization gaps than GNN-based ones. This is particularly obvious when there are 25 axiom combinations: the generalization gap is $66.8\%$ for transformers and $31.6\%$ for GNNs. The test success rate of transformers is $18.2\%$ lower than that of GNNs in this setting. Yet when there are more than 100 axiom combinations in training, transformers always perform better on the test sets, completing $3.8\% - 19.6\%$ more proofs. When the data is diverse, transformers perform better; when it is insufficient, GNNs are better. This might be due to the difference in the inductive bias used by both structures and might explain the choice of neural architectures in deep learning practice.

**Number of Axioms**    Here we investigated how well theorem provers could generalize to test problems that were generated with a different number of axioms than at training time. For instance, let $A$, $B$ and $C$ represent different axioms. Will agents trained on $K2L3$ axiom orders like $[A, B, A]$ and $[C, C, B]$ be able to prove theorems generated with $K3L3$ axiom orders like $[A, B, C]$?

We trained the agents on problems that have the same proof length ($L = 7$) and varying $K$s. The results are on the left of Figure 4. It can be observed from the figure that in general, agents perform the best on the $K$ they were trained on and worse when $K$ shifts away. Transformer-based agents showed better performances in all $K$ and $L$ settings, completing $20.9\%$ more proofs than GNN-based ones on average. The success rates of transformer-based agents drop $5.6\%$ on average when the test $K$ is shifted away by 1 from the training $K$. For GNN-based agents, this averages to $5.1\%$. This shows that their generalization abilities to different number of axioms are similar.

**Proof Length**    We tested the generalization ability of theorem provers over the dimension of proof length of the theorems. To do this, we kept the cardinality of the axiom set to be the same ($K = 3$) and varied the evaluated problems' proof length ($L = 3, 5, 7$). The result is presented on the right of Figure 4. For all of the agents trained, the success rate decreases as the length of the proof increases. This is due to the natural difficulty of completing longer proofs. Observing the figure, we see that the longer the training problems, the less they deteriorate in performance when proofs becomes longer: agents trained on $K3L3$ problems complete $18.8\%$ fewer proofs when $L$ is increased by 1, while ones trained on $K3L7$ complete $5.7\%$ fewer. Furthermore, the performance of transformer-based agents

Table 2: The behavior cloning (BC) agents versus the MCTS-assisted (search) agents. **Left:** The average success rates (in %) of agents with and without MCTS over 1000 test theorems. **Right:** The average length of successful proofs by agents with and without MCTS over 1000 test theorems. $K$ denotes the cardinality of the axiom combination of a proof, $L$ denotes the length of the proof.

| Train | K3L3 | | K3L5 | | K3L7 | | Train | K3L3 | | K3L5 | | K3L7 | |
|---|---|---|---|---|---|---|---|---|---|---|---|---|---|
| Evaluation | BC | Search | BC | Search | BC | Search | Evaluation | BC | Search | BC | Search | BC | Search |
| K3 L3 | 92 | **98** | 91 | 97 | 81 | 96 | K3 L3 | 3.83 | **3.33** | 4.00 | 3.52 | 5.00 | 3.67 |
| K3 L5 | 50 | 64 | 80 | **92** | 70 | **92** | K3 L5 | 7.54 | 6.82 | 6.2 | **5.52** | 6.84 | 5.56 |
| K3 L7 | 25 | 40 | 64 | 78 | 58 | **81** | K3 L7 | 9.05 | 8.54 | 8.01 | 7.53 | 8.39 | **7.50** |
| Average | 56 | 67 | 78 | 89 | 69 | **90** | Average | 6.81 | 6.23 | 6.07 | **5.52** | 6.74 | 5.58 |

decreases by $12.2\%$ when the test proof length increases by 1, compared to $10.7\%$ for GNN-based ones. This suggests that transformers have inferior proof length generalization abilities than GNNs.

## 4.4 GENERALIZING WITH SEARCH

We investigated whether performing search at test time can help agents generalize. Specifically, we investigated the effectiveness of Monte-Carlo Tree Search (MCTS) in finding proofs for unseen theorems with GNN-based agents. We chose GNN-based agents because they are better at out-of-distribution generalization than transformer-based ones. Straightforward application of MCTS is impractical: in our theorem proving environment, the action space can be as large as 1.3M in size (see Appendix H). Hence, it would be infeasible to expand all possible actions when constructing the MCTS trees. Thus, we only performed MCTS over the axiom space (18 distinct axioms in total), and the arguments were proposed by the behavior cloning agents. Following AlphaGo Zero/AlphaZero (Silver et al., 2017; 2018), we trained a value network to estimate the value of a state. The value network is an MLP with two hidden layers of size 256, taking the GNN global representations of graphs as input. It was trained on 1000 episodes of rollouts obtained by the behavior cloning agents, with a learning rate of $3 \cdot 10^{-6}$. We also followed AlphaZero for the choice of the upper confidence bound, and the way that actions are proposed using visit counts. We used 200 simulations for constructing MCTS trees. More details can be found in Appendix F. We took the agents trained on "$K3L3$", "$K3L5$", and "$K3L7$" from section 4.3, and evaluated the agents' performance when boosted by MCTS.

**Generalization** The average success rates on 1000 test theorems are presented on the left in Table 2. We can see that search greatly improved the generalization results. It helped to solve $21\%$ more problems on average for the agent trained on theorem distribution $K3L7$. Remarkably, when evaluating on $K3L7$ theorems, search helped the $K3L3$ agent improve its success rate from $25\%$ to $40\%$: a relative improvement of $60\%$. It is interesting to see the $K3L7$ behavior cloning agent solved $9\%$ fewer problems on average than the $K3L5$ agent. But search brought about much larger improvement to the $K3L7$ agent and helped it to solve the largest proportion of problems on average – $90\%$. This indicates that skills learned through behavior cloning can be better exploited by searching.

The average proof length for 1000 problems is presented on the right in Table 2 (we count those unsolved problem as 15, the step limit of an episode). We can see that by performing search, we are able to discover proofs of length closer to the ground truth proof length. For test theorems requiring 3-step proofs, the $K3L3$ agent was able to prove them in 3.33 steps on average, with a gap of 0.33 steps to the optimal value. Similarly, for test theorems requiring 5-step proofs, the $K3L5$ agent was able to prove them in 5.52 steps on average, with a gap of 0.52 steps; and for theorems requiring 7-step proofs, $K3L7$ agent achieved a gap of 0.5 steps.

## 4.5 DISCUSSION

Experimental results suggested that transformer-based agents can complete more proofs in the IID generalization scenario but have larger out-of-distribution generalization gaps than GNN-based ones. The larger gap may be due to the lack of constraints in the sequence-to-sequence framework, in which the model can propose sequences that are invalid actions, whereas the graph interface constrains the model to propose valid actions only. However, we still see that transformers are able to complete more proofs overall. This shows the superiority of transformers in model capacity when applied to theorem proving. This insight motivates us to explore the possibility of taking the best from both worlds, combining both graph structural information and the strong transformer architecture to improve learning-assisted theorem proving. We leave it for future work.

## 5 CONCLUSION

We addressed the problem of diagnosing the generalization weaknesses in learning-assisted theorem provers. We constructed INT, a synthetic benchmark of inequalities, to analyze the generalization of machine learning methods. We evaluated transformer-based and GNN-based agents and a variation of GNN-based agents with MCTS at test time. Experiments revealed that transformer-based agents generalize better when the IID assumption holds while GNN-based agents generalize better in out-of-distribution scenarios. We also showed that search can boost the generalization ability of agents. We stress that proving theorems in INT is not an end in itself. A hard-coded expert system might perform well on INT but not generalize to real-world mathematical theorems. Therefore, INT should be treated as *instrumental* when diagnosing generalization of agents. The best practice is to use INT in conjunction with real mathematical datasets.

We believe our benchmark can also be of interest to the learning community, facilitating research in studying generalization beyond the IID assumption. The agents' abilities to reason and to go beyond the IID assumption are essential in theorem proving, and studying how to acquire these abilities is at the frontier of learning research. In other domains requiring out-of-distribution generalization, such as making novel dialogs (Chen et al., 2017) or confronting unseen opponents in Starcraft (Vinyals et al., 2019), the requirements for data and computation forbid a generally affordable research environment. The INT benchmark provides practical means of studying out-of-distribution generalization.

## ACKNOWLEDGEMENTS

We thank Jay McClelland, Han Huang and Yuanhao Wang for helpful comments and discussions. We also thank anonymous reviewers for valuable and constructive feedbacks. We are grateful to the Vector Institute for providing computing resources. YW was supported by the Google PhD fellowship. AQJ was supported by a Vector Institute research grant.

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

## APPENDIX A   AXIOM SPECIFICATIONS

| Field axioms | Definition |
|---|---|
| AdditionCommutativity (`AC`) | $\rightarrow a + b = b + a$ |
| AdditionAssociativity (`AA`) | $\rightarrow a + (b + c) = (a + b) + c$ |
| AdditionSimplification (`AS`) | $a = b \rightarrow a + (-b) = 0$ |
| MultiplicatoinCommutativity (`MC`) | $\rightarrow a \cdot b = b \cdot a$ |
| MultiplicationAssociativity (`MA`) | $\rightarrow a \cdot (b \cdot c) = (a \cdot b) \cdot c$ |
| MultiplicationSimplification (`MS`) | $(a \neq 0) \wedge (a = b) \rightarrow 1 = a \cdot \frac{1}{b}$ |
| AdditionMultiplicationLeftDistribution (`AMLD`) | $\rightarrow (a + b) \cdot c = a \cdot c + b \cdot c$ |
| AdditionMultiplicationRightDistribution (`AMRD`) | $\rightarrow a \cdot (b + c) = a \cdot b + a \cdot c$ |
| SquareDefinition (`SD`) | $\rightarrow a^2 = a \cdot a$ |
| MultiplicationOne (`MO`) | $\rightarrow a \cdot 1 = a$ |
| AdditionZero (`AZ`) | $\rightarrow a + 0 = a$ |
| PrincipleOfEquality (`POE`) | $(a = b) \wedge (c = d) \rightarrow a + c = b + d$ |
| EquMoveTerm(Helper axiom) (`EMT`) | $a + b = c \rightarrow a = c + (-b)$ |
| **Ordered field axioms** | Definition |
| All field axioms | |
| SquareGEQZero (`SGEQZ`) | $a = b \rightarrow a \cdot b \geq 0$ |
| EquivalenceImpliesDoubleInequality (`EIDI`) | $a = b \rightarrow (a \geq b) \wedge (a \leq b)$ |
| IneqMoveTerm (`IMT`) | $a + b \geq c \rightarrow a \geq c + (-b)$ |
| FirstPrincipleOfInequality (`FPOI`) | $(a \geq b) \wedge (c \geq d) \rightarrow a + c \geq b + d$ |
| SecondPrincipleOfInequality (`SPOI`) | $(a \geq b) \wedge (c \geq 0) \rightarrow a \cdot c \geq b \cdot c$ |

Table 3

## APPENDIX B   THE MORPH FUNCTION

We detail the morphing of $C$ at each step as follows. For each theorem $a$, we define two symbolic patterns: $\mathcal{L}_a$ and $\mathcal{R}_a$, each represented by an expression (see Appendix C for full details). For example, if $a$ is AdditionCommutativity, we use $\mathcal{L}_a = x_1 + x_2$ to denote any formula that is a sum of two terms ($x_1$ and $x_2$ can be arbitrary terms). We check if one of the nodes in the computation graph of $C$ has the structure defined by $\mathcal{L}_a$. If so, we then transform that node to a formula specified by $\mathcal{R}_a$. For example, if $C$ is $(p + q) + l = (p + (q + l))$, $p + q$ is a node that matches the pattern specified by $\mathcal{L}_a$, in which $x_1 = p$ and $x_2 = q$. Let $\mathcal{R}_a = x_2 + x_1$. We hence transform the node $p + q$ to $q + p$ as specified by $\mathcal{R}_a$. As a result, $C'$ becomes $(q + p) + l = (p + (q + l))$. If there is no node in the computation graph, we morph the core logic statement using the extension function $\mathcal{E}$, defined in Appendix D . We sample nodes in available computation graphs and combine them with $C$, coming up with $C'$ and optionally a non-empty set of new premises $P_{new}$.

---

**Algorithm 2** Theorem Generator (complete)

---

 1: **function** GENERATE_THEOREM(initial conditions $\mathcal{I}$, axiom order $A$)
 2:      Axiom order length $L = \text{len}(A)$.
 3:      Initialize core logic statement $C_0 \sim Uniform(\mathcal{I})$, and the set of premises $P = \{C_0\}$.
 4:      **for** $t \leftarrow 1$ to $L$ **do**
 5:          Get axiom $a_t \leftarrow A[t]$.
 6:          Get new logic statement and premises: $C_t, P_t \leftarrow$ MORPH $(a_t, C_{t-1})$.
 7:          Add new premises to the set of all premises: $P \leftarrow P \cup P_t$.
 8:      **end for**
 9:      **return** $C_L, P$
10: **end function**

     **function** MORPH(axiom $a$, core logic statement $C$)
 2:      Collect $\mathcal{N}_t = \{n|$ n is a node in $C$ and matches the pattern specified by $\mathcal{L}_a\}$
         **if** $\mathcal{N}_t \neq \varnothing$ **then**
 4:          Sample node $n \sim Uniform(\mathcal{N}_t)$.
             Transform $n$ into new node $n'$ using the mapping from $\mathcal{L}_a$ to $\mathcal{R}_a$.
 6:          $C' \leftarrow$ Replace $n$ with $n'$ in the graph of $C$. $P_{new} \leftarrow \varnothing$.
         **else**
 8:          Collect $\mathcal{N}$, the set of all nodes in the graphs.
             Extend $C$ and get the set of premises: $C', P_{new} \leftarrow \mathcal{E}(a, C, \mathcal{N})$.
10:      **end if**
         **return** $C', P_{new}$.
12: **end function**

---

The reasons that we have two sets of rules for morphing are as follow: 1) Transformation rules can only be applied when the axiom will produce an equality, while extension rules can be applied to any axiom. So in order to generate theorems with all the axioms, we need the extension rules. 2) Almost all the extension rules will complicate the core logic statement while none of the transformation rules will. If we only have extension rules, the goal generated can be very complex even the proof is of moderate length. In order to generate compact theorems (goal not too complicated) with long proofs, the transformation rules are preferred. Therefore we only apply extension rules when transformation rules are not applicable.

## APPENDIX C    TRANSFORMATION RULES

The implementations of the transformation rules $\mathcal{L}$ and $\mathcal{R}$.

| Axiom ($a$) | $\mathcal{L}_a$ | $\mathcal{R}_a$ |
|---|---|---|
| AdditionCommutativity | $x_1 + x_2$ | $x_2 + x_1$ |
| AdditionAssociativity | $x_1 + (x_2 + x_3)$ | $(x_1 + x_2) + x_3$ |
| AdditionSimplification | $x_1 + (-x_1)$ | $0$ |
| MultiplicatoinCommutativity | $x_1 \cdot x_2$ | $x_2 \cdot x_1$ |
| MultiplicationAssociativity | $x_1 \cdot (x_2 \cdot x_3)$ | $(x_1 \cdot x_2) \cdot x_3$ |
| MultiplicationSimplification | $x_1 \cdot \frac{1}{x_1}$ | $1$ |
| AdditionMultiplicationLeftDistribution | $(x_1 + x_2) \cdot x_3$ | $x_1 \cdot x_3 + x_2 \cdot x_3$ |
| AdditionMultiplicationRightDistribution | $x_1 \cdot (x_2 + x_3)$ | $x_1 \cdot x_2 + x_1 \cdot x_3$ |
| SquareDefinition | $x_1^2$ | $x_1 \cdot x_1$ |
| MultiplicationOne | $x_1 \cdot 1$ or $1 \cdot x_1$ | $x_1$ |
| AdditionZero | $x_1 + 0$ or $0 + x_1$ | $x_1$ |
| SquareGEQZero | NA | NA |
| PrincipleOfEquality | NA | NA |
| EquMoveTerm | NA | NA |
| EquivalenceImpliesDoubleInequality | NA | NA |
| IneqMoveTerm | NA | NA |
| FirstPrincipleOfInequality | NA | NA |
| SecondPrincipleOfInequality | NA | NA |

Table 4

## APPENDIX D   EXTENSION FUNCTION

For these axioms, the core logic statement $C$ needs to be of the form LHS($C$) = RHS($C$).

| Axiom ($a$) | Extension function $\mathcal{E}(C, a, \mathcal{N})$ |
|---|---|
| AdditionCommutativity | Sample node $n \sim$ `Uniform`($\mathcal{N}$)
**return** RHS($C$) $+n = n+$ LHS($C$), $\varnothing$ |
| AdditionAssociativity | Sample nodes $n_1, n_2 \sim$ `Uniform`($\mathcal{N}$)
**return** RHS($C$)$+(n_1 + n_2) =$LHS($C$)$+n_1 + n_2$, $\varnothing$ |
| AdditionSimplification | **return** $0 =$LHS($C$)$+(-$RHS($C$))$, $\varnothing$ |
| MultiplicatoinCommutativity | Sample node $n \sim$ `Uniform`($\mathcal{N}$)
**return** RHS($C$)$\cdot n = n\cdot$LHS($C$), $\varnothing$ |
| MultiplicationAssociativity | Sample nodes $n_1, n_2 \sim$ `Uniform`($\mathcal{N}$)
**return** RHS($C$)$\cdot(n_1 \cdot n_2) =$LHS($C$)$\cdot n_1 \cdot n_2$, $\varnothing$ |
| MultiplicationSimplification | **return** $1 =$LHS($C$)$\cdot \frac{1}{\text{RHS}(C)}$, $\varnothing$ |
| AdditionMultiplicationLeftDistribution | Sample nodes $n_1, n_2 \sim$ `Uniform`($\mathcal{N}$)
**return** $(n_1 + n_2) \cdot$ RHS($C$) $=$
$\quad n_1 \cdot$ LHS($C$) $+ n_2 \cdot$ LHS($C$), $\varnothing$ |
| AdditionMultiplicationRightDistribution | Sample nodes $n_1, n_2 \sim$ `Uniform`($\mathcal{N}$)
**return** RHS($C$) $\cdot (n_1 + n_2) =$
$\quad$ LHS($C$) $\cdot n_1 +$ LHS($C$) $\cdot n_2$, $\varnothing$ |
| SquareDefinition | **return** LHS($C$) $\cdot$ RHS($C$) $=$ LHS($C$)$^2$, $\varnothing$ |
| MultiplicationOne | **return** `Uniform`( {LHS($C$) $\cdot 1 =$ RHS($C$),
$\quad\quad\quad 1 \cdot$ LHS($C$) $=$ RHS($C$) } ), $\varnothing$ |
| AdditionZero | **return** `Uniform`( {LHS($C$) $+ 0 =$ RHS($C$),
$\quad\quad\quad 0 +$ LHS($C$) $=$ RHS($C$) } ), $\varnothing$ |
| SquareGEQZero | **return** LHS($C$) $\cdot$ RHS($C$) $\geq 0$, $\varnothing$ |
| PrincipleOfEquality | Sample nodes $n_1, n_2 \sim \mathcal{N}$, where $n_1 = n_2$
**return** LHS($C$) $+ n_1 =$ RHS($C$) $+ n_2$, $\{n_1 = n_2\}$ |
| EquMoveTerm | Only execute when LHS($C$) is of the form $x + y$
**return** $x =$ RHS($C$) $+ (-y)$, $\varnothing$ |
| EquivalenceImpliesDoubleInequality | **return** LHS($C$) $\geq$ RHS($C$), $\varnothing$ |

Table 5

For these axioms, the core logic statement $C$ needs to be of the form $\text{LHS}(C) \geq \text{RHS}(C)$.

| Axiom $(a)$ | Extension function $\mathcal{E}(C, a, \mathcal{N})$ |
|---|---|
| IneqMoveTerm | Only execute when $\text{LHS}(C)$ is of the form $x + y$
**return** $x \geq \text{RHS}(C) + (-y), \varnothing$ |
| FirstPrincipleOfInequality | Sample nodes $n_1, n_2 \sim \mathcal{N}$, where $n_1 \geq n_2$
**return** $\text{LHS}(C) + n_1 \geq \text{RHS}(C) + n_2, \{n_1 \geq n_2\}$ |
| SecondPrincipleOfInequality | Sample node $n \sim \mathcal{N}$, where $n \geq 0$
**return** $\text{LHS}(C) \cdot n \geq \text{RHS}(C) \cdot n, \{n \geq 0\}$ |

Table 6

## APPENDIX E    DATASET STATISTICS

### APPENDIX E.1    THEOREM LENGTH

We compare the length of the theorems generated in characters and plot their distributions in Figure 5. The length of the theorem in characters is a measure for how complicated it is. As is expected, the more complicated the theorem is, the longer the proof(bigger $L$). It is also worth noting that as $L$ becomes bigger, the distribution of theorem length becomes less concentrated. This is likely a consequence of a more spread-out theorem length range.

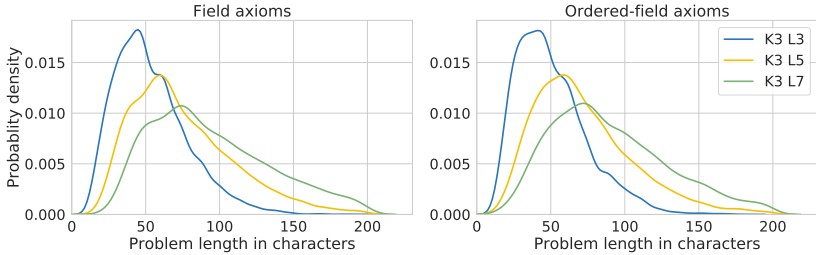

Figure 5: The distribution of theorem length in characters for field axioms(left) and ordered-field axioms(right) generated with parameters $K3L3$, $K3L5$, and $K3L7$. As the length of the proof is increased, so is the number of characters in the theorem, while the distribution of latter is less concentrated.

## APPENDIX E.2    AXIOM DISTRIBUTIONS

The frequency at which each axiom is applied influences the distribution of theorems our generator is able to produce. In Figure 6, we present the proportions of axioms that are applied in generating 10,000 theorems. Their frequencies are a measure of how easy it is to satisfy the conditions to apply them. For the field axioms, the PrincipleOfEquality axiom is the most frequently used(9.30%) and the EquMoveTerm axiom is the most rarely used(2.38%). EquMoveTerm has a strict condition for application: the left hand side of the core logic statement has to be of the form $x + y$, therefore not frequently applied. For the ordered-field axioms, the EquivalenceImpliesDoubleInequality axiom is the most frequently used(10.18%). Since we start with a trivial equality in generation and want to end up with an inequality, a transition from equality to inequality is needed. Among the ways of transitioning, this conditions to apply this axiom is easiest to satisfy. Its popularity is followed by the group of Field axioms, from MultiplicationCommutativity(4.69%) to AdditionAssociativity(5.98%). The rest are ordered-field axioms which define the properties of inequalities, proportions ranging from IneqMoveTerm(1.14%) to FirstPrincipleOfInequality(5.74%).

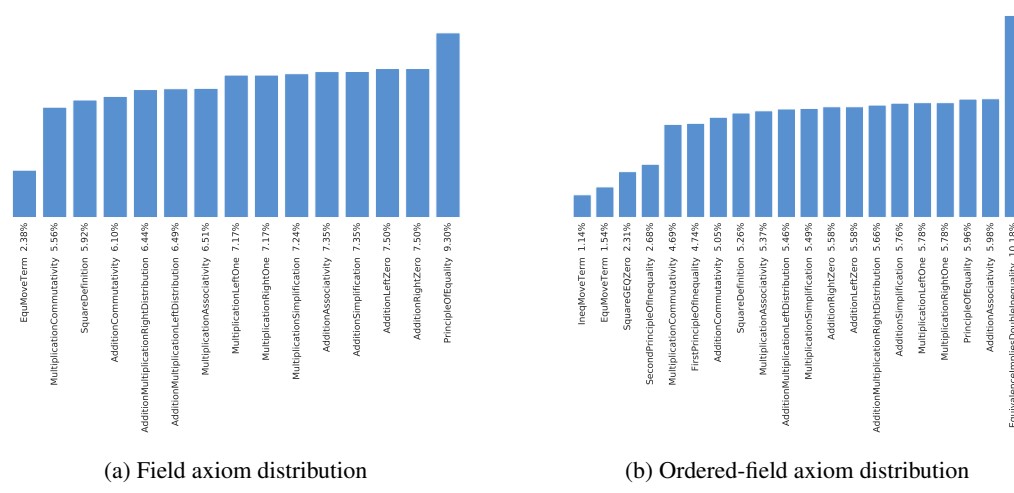

(a) Field axiom distribution    (b) Ordered-field axiom distribution

Figure 6

APPENDIX E.3   NUMBER OF NODES

Since an action in the MDP consists of an axiom and a list of nodes as its arguments and the number of axioms is fixed, the number of nodes available determines the size the action space. Therefore it is interesting to investigate how many nodes are available in a proof. In Figure 7 we present the average number of nodes in proofs of different length. It can be told from the figure that the longer the proofs, the more nodes there will be, as expected. Comparing the axiom sets used, we find that the average number of nodes for ordered-field axioms is larger than that of field axioms. This is likely the consequence of ordered-field axioms, in generation, being more capable of producing new premises(e.g. First Principle of Inequality will produce an inequality premise(see Table 6), thus adding more nodes in the graphs).

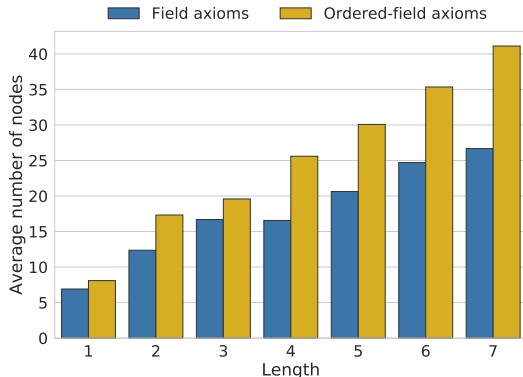

Figure 7

## APPENDIX F    MORE EXPERIMENTAL DETAILS FOR GENERALIZATION WITH SEARCH

We give more experimental details for the use of MCTS. Following (Silver et al., 2017), in the selection step of the MCTS tree construction, we use the following formula to select the next action,

$$a^* = \mathrm{argmax}_a \left( Q(s,a) + c_{\mathrm{puct}} P(s,a) \frac{\sqrt{\sum_b N(s,b)}}{1 + N(s,a)} \right),$$

where $Q(s,a)$ represents the action value function, $N(s,a)$ denotes the visit counts, $P(s,a)$ is the prior probability, and $c_{\mathrm{puct}}$ is a constant hyperparameter. In all of our experiments, we used the behavior cloning policy for computing $P(s,a)$, and we used $c_{\mathrm{puct}} = 1$. After the MCTS tree is built, the action is sampled from the policy distribution $\pi(a|s) = N(s,a)^{\frac{1}{\tau}}$, where $\tau$ is a hyperparameter and was chosen as 1 in our experiments.

# APPENDIX G  MORE TRAINING AND EVALUATION RESULTS

## APPENDIX G.1  LEARNING CURVES OF GNN-BASED AGENTS

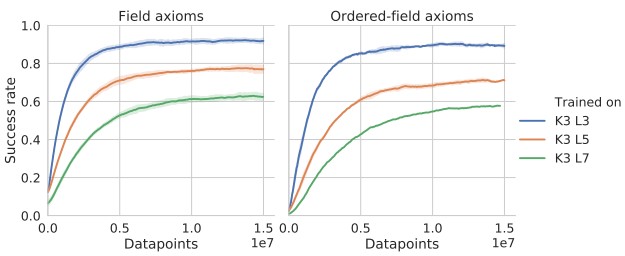

Figure 8: Proof success rates for field axioms(left) and ordered-field axioms(right) of GNN-based agents trained on different $K$ and $L$ parameters. We keep the $K$ the same and vary the $L$. The agents converge slower and to a lower success rate when the proof length is increased. Also, the agents on field axioms are easier to train than those on ordered-field axioms.

## APPENDIX G.2  PERFORMANCE VARIATION OF TRAINED AGENTS

To verify that the experimental results are statistically significant, we ran the experiments on proof length generalization in subsection 4.3 with 5 random seeds and tabled the results.

Table 7: Success rates of agents trained and tested on problems of different parameters (mean $\pm$ std) in percentage.

| Transformers | Tested on | K3 L3 | K3 L5 | K3 L7 |
|---|---|---|---|---|
|  | K3 L3 | $97.6 \pm 0.9$ | $31.5 \pm 1.6$ | $10.9 \pm 1.0$ |
| Trained on | K3 L5 | $97.2 \pm 0.7$ | $88.3 \pm 1.2$ | $59.5 \pm 1.6$ |
|  | K3 L7 | $96.6 \pm 1.2$ | $87.0 \pm 1.6$ | $75.1 \pm 1.2$ |
| GNNs | Tested on | K3 L3 | K3 L5 | K3 L7 |
|  | K3 L3 | $91.5 \pm 0.5$ | $45.6 \pm 1.7$ | $16.5 \pm 0.8$ |
| Trained on | K3 L5 | $86.4 \pm 0.9$ | $77.8 \pm 0.9$ | $58.4 \pm 1.5$ |
|  | K3 L7 | $82.0 \pm 1.3$ | $71.4 \pm 1.1$ | $56.5 \pm 1.5$ |

## APPENDIX G.3  GNN-BASED AGENTS ON IID GENERALIZATION

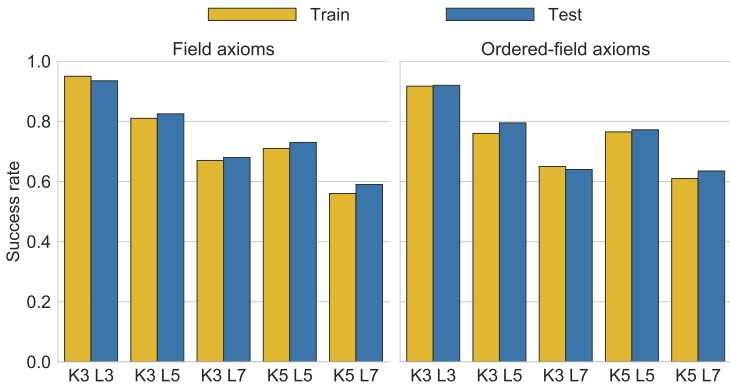

Figure 9: Proof success rates on problems generated with different $K$ and $L$ parameters ($K$ denotes the cardinality of the axiom combination of a proof, $L$ denotes the length of the proof). When the IID assumption holds, the success rate decreases as the two generation parameters $K$ and $L$ are increased.

APPENDIX G.4   GNN-BASED AGENTS ON INITIAL CONDITION GENERALIZATION

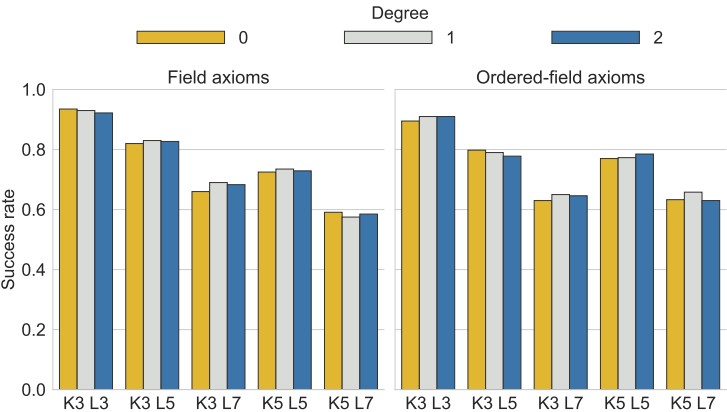

Figure 10: Proof success rates on problems generated with different $K$ and $L$ parameters ($K$ denotes the cardinality of the axiom combination of a proof, $L$ denotes the length of the proof). When generalizing to different initial conditions, there are no obvious trends as to how the proof success rate changes as the degree of the initial entities is varied.

APPENDIX G.5    FULL RESULTS ON AXIOM ORDERS AND COMBINATIONS GENERALIZATION

Table 8: **Top:** Proof success rates (in %) of agents trained on different numbers of axiom orders. **Bottom:** Proof success rates (in %) of agents trained on different numbers of axiom combinations. $K$ denotes the cardinality of the axiom combination of a proof, $L$ denotes the length of the proof.

| Architecture | Axiom orders | 100 | | 500 | | 2000 | | 5000 | |
|---|---|---|---|---|---|---|---|---|---|
| | | Train | Test | Train | Test | Train | Test | Train | Test |
| Transformer | K3 L3 | 98.4 | 32.6 | 99.5 | 90.0 | 98.8 | 98.7 | 97.6 | 97.6 |
| | K3 L5 | 95.3 | 6.3 | 94.0 | 56.3 | 94.0 | 94.9 | 96.5 | 94.9 |
| | K3 L7 | 87.8 | 3.8 | 88.0 | 46.4 | 88.3 | 77.5 | 88.4 | 85.5 |
| | K5 L5 | 94.7 | 5.6 | 97.0 | 72.9 | 97.4 | 93.1 | 97.5 | 96.9 |
| | K5 L7 | 89.7 | 1.8 | 88.6 | 48.6 | 89.3 | 75.2 | 88.6 | 84.0 |
| | **Average** | 93.2 | 10.0 | 93.4 | 62.8 | 93.6 | 87.9 | 93.7 | 91.8 |
| GNN | K3 L3 | 84.3 | 38.6 | 94.4 | 73.9 | 93.7 | 89.0 | 90.5 | 92.3 |
| | K3 L5 | 92.7 | 17.1 | 86.3 | 60.0 | 84.4 | 72.9 | 77.7 | 77.1 |
| | K3 L7 | 82.4 | 14.1 | 82.4 | 33.8 | 68.6 | 57.7 | 70.2 | 63.5 |
| | K5 L5 | 91.0 | 23.0 | 89.7 | 61.2 | 81.8 | 75.0 | 78.3 | 80.8 |
| | K5 L7 | 87.5 | 12.9 | 80.2 | 39.0 | 66.5 | 57.4 | 61.6 | 60.0 |
| | **Average** | 87.6 | 21.1 | 86.6 | 53.6 | 79.0 | 70.4 | 75.7 | 74.7 |

| Architecture | Axiom combos | 25 | | 100 | | 200 | | 300 | |
|---|---|---|---|---|---|---|---|---|---|
| | | Train | Test | Train | Test | Train | Test | Train | Test |
| Transformer | K3 L3 | 99.2 | 34.1 | 99.0 | 72.8 | 99.5 | 96.1 | 98.6 | 98.2 |
| | K3 L5 | 97.8 | 29.3 | 98.6 | 66.3 | 97.5 | 89.5 | 94.3 | 90.4 |
| | K3 L7 | 93.6 | 25.0 | 91.9 | 55.9 | 91.5 | 80.0 | 91.9 | 85.9 |
| | K5 L5 | 98.5 | 27.4 | 98.4 | 87.6 | 97.0 | 93.6 | 97.3 | 94.9 |
| | K5 L7 | 91.2 | 30.5 | 92.2 | 76.3 | 91.7 | 82.9 | 90.0 | 87.0 |
| | **Average** | 96.1 | 29.3 | 96.0 | 71.8 | 95.4 | 88.4 | 94.4 | 91.3 |
| GNN | K3 L3 | 96.3 | 61.6 | 96.0 | 90.1 | 92.7 | 91.2 | 95.3 | 92.0 |
| | K3 L5 | 82.1 | 43.4 | 80.3 | 68.9 | 78.5 | 74.9 | 76.5 | 76.1 |
| | K3 L7 | 72.1 | 34.3 | 68.1 | 57.2 | 62.3 | 63.7 | 62.5 | 62.0 |
| | K5 L5 | 77.8 | 61.6 | 78.9 | 71.0 | 74.5 | 78.4 | 72.8 | 74.9 |
| | K5 L7 | 67.2 | 36.8 | 59.7 | 52.7 | 54.9 | 54.0 | 56.7 | 54.5 |
| | **Average** | 79.1 | 47.5 | 76.6 | 68.0 | 72.6 | 72.4 | 72.8 | 71.9 |

### APPENDIX G.6    GNN-BASED AGENTS ON AXIOM NUMBER GENERALIZATION

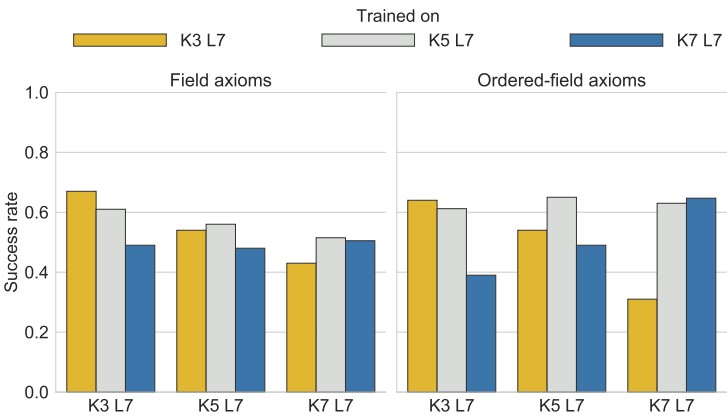

Figure 11: Proof success rates on problems generated with different parameters (($K$ denotes the cardinality of the axiom combination of a proof, $L$ denotes the length of the proof). We keep parameter $L$ the same and vary parameter $K$. The success rate is likely to decrease when an agent is evaluated on problems that have different $K$ than the problems it is trained on.

### APPENDIX G.7    GNN-BASED AGENTS ON PROOF LENGTH GENERALIZATION

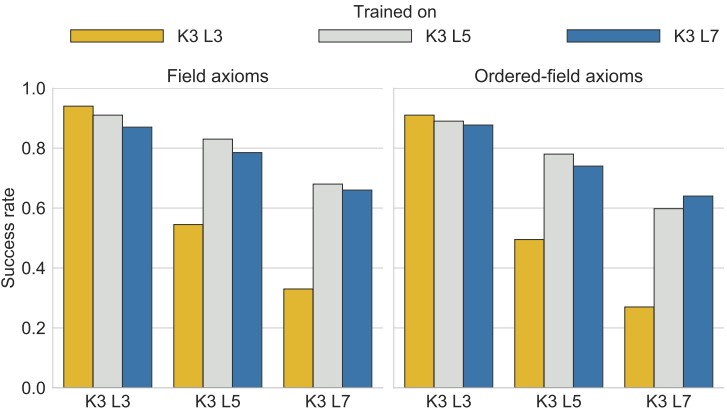

Figure 12: Proof success rates on problems generated with different parameters (($K$ denotes the cardinality of the axiom combination of a proof, $L$ denotes the length of the proof). We keep parameter $K$ the same and vary parameter $L$. For all agents, the proof success rate is lower on theorems that require longer proofs. The best-performing agent for problems of a given length is usually the agent trained on problems of the same length.

## APPENDIX H    THEOREM PROVING AS A MARKOV DECISION PROCESS (MDP)

We model theorem proving as a Markov Decision Process. A state $s$ in the MDP is the proof state maintained by the assistant, namely, the goal, the premises and the proven facts, represented by computation graphs. An action $a$ is a tuple of an axiom and a sequence of arguments. We denote the axiom space as $\mathcal{X}$ and the argument space, the set of all the nodes in available computation graphs, as $\mathcal{N}$. The maximum number of arguments for one axiom within our axiomizations is 3, therefore the action space is $\mathcal{A} = \mathcal{X} \times \mathcal{N}^3$. The assistant ignores redundant arguments if fewer than 3 are needed for the axiom considered. We show in Appendix E.3 the distribution of the number of nodes for proofs of different length. The size of the discrete action space can be as large as $18 \times 42^3 \approx 1.33 \times 10^6$. The deterministic state transition function $P(s, a)$ is implicitly determined by the proof assistant. When the proof assistant deems the proof complete and the theorem proven, the episode terminates and a reward of one is given. Otherwise, the reward is zero at each step. When the step limit for a proof is exhausted, the episode terminates with a reward of zero. For experiments in this paper, we used a step limit of 15.

# APPENDIX I   EXAMPLE PROBLEMS

## Equality theorems

**Theorem 1**
Goal: $((0 \cdot 1) \cdot ((-(a^2)) \cdot c)) = (((-(a^2)) \cdot ((a \cdot a) + (-(a^2)))) \cdot c)$

**Theorem 2**
Goal: $(((((0 + c) + a) \cdot a) \cdot 1) \cdot (b \cdot (0 + c))) = ((((c \cdot a) + (a \cdot a)) \cdot (0 + c)) \cdot b)$

**Theorem 3**
Goal: $0 = ((((c + 0) \cdot (a + a)) \cdot (\frac{1}{((c \cdot a) + (c \cdot a))})) + (-(0 + 1)))$

**Theorem 4**
Premises: $(b + d) = b$
Goal: $(1 + (-((b + b) \cdot (\frac{1}{((b + (b + d)) \cdot 1)})))) = (0 + 0)$

**Theorem 5**
Premises: $(a + d) = b$
Goal: $1 = (((d \cdot ((a + d) + ((c + (a + d)) + 0))) \cdot ((d \cdot (a + d)) + (d \cdot (c + b)))) \cdot (\frac{1}{((d \cdot ((a + d) + ((c + (a + d)) + 0)))^2)}))$

**Theorem 6**
Premises: $((b \cdot b) + d) = (b \cdot b)$
Goal: $(0 + ((b \cdot b) + d)) = (((1 \cdot ((b + b) \cdot b)) + (-(((b \cdot b) + (b \cdot b)) \cdot 1))) + (b \cdot b))$

**Theorem 7**
Goal: $((a \cdot (a + 0)) + ((-(0 + a)) \cdot (a + 0))) = ((a \cdot 0) + (0 \cdot 0))$

**Theorem 8**
Goal: $(((c \cdot c) + c) \cdot ((c^2) \cdot 1)) = (((c \cdot c) \cdot (0 + (c \cdot c))) + (c \cdot (0 + (c \cdot c))))$

**Theorem 9**
Goal: $1 = ((((a \cdot c) + ((b \cdot (a \cdot b)) \cdot c)) \cdot (a + (a \cdot c))) \cdot (\frac{1}{((((a + ((b \cdot a) \cdot b)) \cdot c) \cdot (a \cdot c)) + (((a + ((b \cdot a) \cdot b)) \cdot c) \cdot a))}))$

**Theorem 10**
Goal: $((((b \cdot c) + (c \cdot c)) + (-(0 + ((b + c) \cdot c)))) \cdot (c \cdot c)) = ((c^2) \cdot 0)$

**Theorem 11**
Goal: $(1 \cdot (b + a)) = ((0 + (a + b)) + 0)$

**Theorem 12**
Goal: $(((-c) \cdot (-c)) + (((-c) \cdot c) + ((-c) \cdot (-c)))) = (((-c) \cdot (-c)) + (0 \cdot (-c)))$

**Theorem 13**
Goal: $(((a^2) \cdot (a \cdot (a + 0))) + (a \cdot (a \cdot (a + 0)))) = ((((a^2) \cdot (a^2)) + (a \cdot (a^2))) + 0)$

**Theorem 14**
Goal: $((((b \cdot 1) \cdot (a \cdot c)) \cdot (b \cdot a)) + (((b \cdot 1) \cdot (a \cdot c)) \cdot (b \cdot a))) = (((((b \cdot a) \cdot c) \cdot (b \cdot a)) + (((b \cdot a) \cdot c) \cdot (b \cdot a))) \cdot 1)$

**Theorem 15**
Goal: $1 = ((\frac{1}{((\frac{1}{(b + 0)}) \cdot b)}) \cdot 1)$

**Theorem 16**

Goal: $0 = ((0 + (-((a \cdot b) + (-(b \cdot a))))) + (-(0 \cdot 1)))$

**Theorem 17**
Premises: $(a + d) = c$; $((b + c) + e) = (a + d)$
Goal: $(((b \cdot d) + (b \cdot (b + (a + d)))) + ((b + c) + e)) = ((((b \cdot d) + (b \cdot (b + c))) \cdot 1) + (a + d))$

**Theorem 18**
Goal: $((((\frac{1}{b}) \cdot b) \cdot b) \cdot 1) = ((b \cdot 1) \cdot 1)$

**Theorem 19**
Goal: $(((1 \cdot (b \cdot (c + a))) + (b \cdot a)) + 1) = (1 \cdot ((1 \cdot ((b \cdot c) + (b \cdot a))) + ((b \cdot a) + 1)))$

**Theorem 20**
Premises: $(b + d) = c$; $((1 \cdot a) + e) = a$
Goal: $(((a + (b + d)) \cdot (\frac{1}{((1 \cdot a) + c)})) + ((1 \cdot a) + e)) = ((1 \cdot 1) + a)$

**Theorem 21**
Goal: $((((c^2) \cdot ((c^2) \cdot c)) + (-(((c \cdot c) \cdot (c^2)) \cdot c))) + (b + b)) = ((1 \cdot ((0 + b) + b)) + (-0))$

**Theorem 22**
Premises: $(b + d) = (a \cdot b)$
Goal: $(1 \cdot ((((c+c) \cdot (((a \cdot b) \cdot c) + (c+c))) + ((c+c) \cdot (c+c))) + (a \cdot b))) = (((((c+c) \cdot (((a \cdot (b \cdot c)) + c) + c) + (c+c))) + (b+d)) \cdot 1) + 0)$

**Theorem 23**
Premises: $((0 \cdot 1) + d) = (1 \cdot 0)$
Goal: $(((((a + (0 \cdot 1)) \cdot (1 \cdot 0)) + (-b)) + (1 \cdot 0)) + (1 \cdot 0)) = (((((a \cdot (1 \cdot 0)) + ((b + (-b)) \cdot (1 \cdot 0))) + ((-b) + (1 \cdot 0))) + ((0 \cdot 1) + d)) + 0)$

**Theorem 24**
Premises: $(a + d) = (1 + c)$
Goal: $(((((1 \cdot b) + (c \cdot b)) + (1 + c))^2) \cdot ((1 + c) \cdot b)) = ((((((1 \cdot b) + (c \cdot b)) + (1 + c)) \cdot (((b \cdot (1 + (1 \cdot c))) + (a + d)) \cdot 1)) \cdot (1 + c)) \cdot b)$

**Theorem 25**
Premises: $(a + d) = (b \cdot 1)$
Goal: $0 = ((b + (a + d)) + (-((b \cdot 1) + (b \cdot 1))))$

**Theorem 26**
Premises: $(c + d) = a$
Goal: $(0 + ((((a + a) \cdot 1) + a) \cdot 1)) = (1 \cdot ((1 \cdot ((a + (c + d)) + 0)) + (1 \cdot a)))$

**Theorem 27**
Premises: $(c + d) = (b + c)$
Goal: $(1 \cdot ((((b + c) \cdot c) + (b \cdot (b + c))) + (c + d))) = ((((b + c)^2) + (b + c)) \cdot 1)$

**Theorem 28**
Premises: $((1 \cdot b) + d) = b$
Goal: $(((((1 \cdot b) + b) \cdot (a \cdot 1)) \cdot (((b + ((1 \cdot b) + d)) \cdot a) \cdot 1)) + 0) = ((((b + (1 \cdot b)) \cdot (a \cdot 1))^2) \cdot 1)$

**Theorem 29**
Goal: $(((b \cdot 1) + 0) \cdot (1 \cdot 0)) = (((b \cdot 1) \cdot ((-(0 + b)) + (1 \cdot b))) + (0 \cdot ((-(0 + b)) + (1 \cdot b))))$

**Theorem 30**

Goal: $(1 \cdot 1) = (((((a \cdot (c + c)) + 0) \cdot (b \cdot (c + c))) \cdot (\frac{1}{((((a \cdot c) + (a \cdot c)) \cdot b) \cdot (c + c))})) + 0)$

**Theorem 31**
Goal: $((1 \cdot (b \cdot b)) \cdot b) = (1 \cdot (0 + (((0 + b) \cdot b) \cdot b)))$

**Theorem 32**
Goal: $(((c \cdot (c \cdot 1)) + 0) \cdot 1) = (((c \cdot c) + 0) \cdot 1)$

**Theorem 33**
Goal: $1 = (1 \cdot (\frac{1}{((1 + 0) \cdot (\frac{1}{((b \cdot (\frac{1}{b})) + 0)}))}))$

**Theorem 34**
Goal: $(((((((c + a) \cdot a) \cdot (c + a)) \cdot c) \cdot (a + c)) \cdot (c + a)) \cdot (c + a)) = (((((((a + c) \cdot (c + a)) \cdot a) \cdot c) \cdot (a + c)) \cdot (c + a)) \cdot (c + a))$

**Theorem 35**
Goal: $0 = ((-(1 \cdot 0)) + ((-(c + c)) + ((1 \cdot c) + c)))$

**Theorem 36**
Goal: $1 = (1 \cdot (\frac{1}{(a \cdot (\frac{1}{(((a + c) + a) + (-(c + a)))}))}))$

**Theorem 37**
Premises: $(a + d) = a$; $((\frac{1}{c}) + e) = b$
Goal: $(((1 \cdot (1 \cdot (\frac{1}{(c \cdot (\frac{1}{c}))}))) + a) + b) = (1 \cdot (((1 \cdot 1) + (a + d)) + ((\frac{1}{c}) + e)))$

**Theorem 38**
Goal: $0 = ((b \cdot (b + (-b))) + (-(((0 + 0) \cdot b) + 0)))$

**Theorem 39**
Goal: $(((1 \cdot c) + (-(1 \cdot (c \cdot 1)))) \cdot 1) = ((0 \cdot 1) \cdot 1)$

**Theorem 40**
Goal: $((a + b) \cdot (1 \cdot ((b \cdot c) + (c \cdot c)))) = ((a \cdot ((c \cdot c) + (b \cdot c))) + (b \cdot ((c \cdot c) + (b \cdot c))))$

**Theorem 41**
Goal: $(0 + ((0 + ((c + c) \cdot c)) \cdot (a \cdot b))) = (0 + ((((c \cdot c) \cdot a) + ((c \cdot c) \cdot a)) \cdot b))$

**Theorem 42**
Premises: $(0 + d) = 1$
Goal: $((((1 \cdot 0) + (a + (a \cdot 1))) + 0) + d) = (((((1 \cdot a) + (-(a \cdot 1))) + a) + (a \cdot 1)) + 1)$

**Theorem 43**
Premises: $(b + d) = 0$
Goal: $0 = ((((((0 + b) \cdot 0) + ((0 + b) \cdot b)) \cdot 1) + 0) + (-((((b \cdot 0) + (b \cdot b)) + (b + d)) \cdot 1)))$

**Theorem 44**
Goal: $((0 + c) \cdot ((-c) + (((c \cdot 1) + 0) + (-c)))) = (((0 + c) \cdot (-c)) + ((0 + c) \cdot 0))$

**Theorem 45**
Goal: $0 = (0 + (-(((0 \cdot 0) + (a \cdot 0)) + (-((((((a \cdot b) + (a \cdot b)) + ((b + b) + b)) + (-(((a \cdot (b + b)) + (b + b)) + b))) + a) \cdot 0) \cdot 1)))))$

**Theorem 46**

Premises: $((a + b) + d) = (a + b)$; $(b + e) = a$
Goal: $(a \cdot a) = (1 \cdot (a \cdot a))$

**Theorem 47**
Premises: $(c + d) = c$
Goal: $((b \cdot (1 + 0)) + (b \cdot (c + d))) = (0 + (b \cdot ((0 + (1 \cdot (\frac{1}{((b+(c+d)) \cdot (\frac{1}{(b+c)}))}))) + (c + d))))$

**Theorem 48**
Goal: $((b + ((((a + a) \cdot 1) \cdot a) + 0)) \cdot a) = ((b \cdot a) + ((((a \cdot a) \cdot 1) + (a \cdot (a \cdot 1))) \cdot a))$

**Theorem 49**
Goal: $(((1 + b) \cdot (((a \cdot ((c \cdot 1) + (c^2))) + 1) + b)) + ((1 + b) \cdot (((a \cdot ((c \cdot 1) + (c^2))) + 1) + b))) = ((((1 + b) + (1 + b)) \cdot ((((a \cdot (c \cdot 1)) + (a \cdot (c \cdot (c \cdot 1)))) + (1 + b)) \cdot 1)) \cdot 1)$

**Theorem 50**
Goal: $0 = (((((0 + (((c \cdot c) + (c \cdot c)) + ((c + c) + (c \cdot c)))) + 0) + c) \cdot a) + (-(((0 + ((((c + c) \cdot c) + (c + c)) + (c \cdot c))) \cdot a) + (c \cdot a))))$

# Inequality theorems

**Theorem 1**
Premises: $(1 + d) \geq 0$; $(b + e) \geq 0$
Goal: $((((1 + 1) \cdot (a \cdot (\frac{1}{a}))) \cdot (1 + d)) + (b + e)) \geq ((((1 \cdot 1) + (1 \cdot 1)) \cdot (1 + d)) + 0)$

**Theorem 2**
Goal: $(b^2) \geq (0 + (b \cdot (1 \cdot b)))$

**Theorem 3**
Premises: $((c + 0) + d) \geq 0$; $(d + e) \geq b$
Goal: $((c \cdot ((c + 0) + d)) + (d + e)) \geq (((0 + c) \cdot ((c + 0) + d)) + b)$

**Theorem 4**
Goal: $(b + 0) \geq ((((0 + b) + c) + c) + (-(c + c)))$

**Theorem 5**
Premises: $(1 + d) \geq 0$
Goal: $(((((c \cdot c) + c) + a) \cdot (1 + d)) \geq ((((c^2) + (c + a)) \cdot 1) \cdot (1 + d))$

**Theorem 6**
Premises: $(b + d) = b$
Goal: $1 \geq ((((a + b) + (-(b + d))) \cdot ((a + b) + b)) \cdot (\frac{1}{((a \cdot (a+b)) + (a \cdot b))}))$

**Theorem 7**
Premises: $((0 + a) + d) = 0$
Goal: $(((0 + a) \cdot a) + ((0 + a) + d)) \geq ((a^2) + 0)$

**Theorem 8**
Premises: $(b + d) = a$
Goal: $((c \cdot b) + (b \cdot b)) \geq (1 \cdot ((((c + a) + (-(b + d))) + b) \cdot b))$

**Theorem 9**
Goal: $(1 \cdot ((((b \cdot (\frac{1}{b})) \cdot a) \cdot (1 \cdot 1)) + a)) \geq ((1 \cdot ((1 \cdot 1) \cdot (a \cdot (1 \cdot 1)))) + (1 \cdot a))$

**Theorem 10**
Premises: $(c + d) \geq 0$
Goal: $(b \cdot (c + d)) \geq ((((b + b) + 0) + (-b)) \cdot (c + d))$

**Theorem 11**
Goal: $(((b + 0) + (b + c)) + 0) \geq (((b + b) + c) + 0)$

**Theorem 12**
Goal: $((c \cdot (c + 0)) + 0) \geq ((c^2) + 0)$

**Theorem 13**
Goal: $(1 \cdot (b \cdot 1)) \geq ((1 \cdot b) \cdot 1)$

**Theorem 14**
Goal: $1 \geq ((((b \cdot (\frac{1}{b})) + (\frac{1}{b})) + 1) \cdot (\frac{1}{((1 + (\frac{1}{b})) + 1)}))$

**Theorem 15**
Goal: $1 \geq ((\frac{1}{((c \cdot a) \cdot (\frac{1}{(a \cdot c)}))}) \cdot 1)$

**Theorem 16**
Goal: $((c \cdot (a \cdot a)) + (((a \cdot a) + (c \cdot a)) \cdot (a \cdot a))) \geq (0 + ((c + (0 + ((a + c) \cdot a))) \cdot (a \cdot a)))$

**Theorem 17**
Goal: $(((c \cdot b) + a) \cdot ((c \cdot b) + (c \cdot b))) \geq ((a \cdot ((c \cdot b) + (c \cdot b))) + ((c \cdot b) \cdot ((c \cdot b) + (c \cdot b))))$

**Theorem 18**
Goal: $((a \cdot b) \cdot 1) \geq ((((a \cdot 1) \cdot b) \cdot 1) \cdot 1)$

**Theorem 19**
Goal: $a \geq ((a + c) + (-c))$

**Theorem 20**
Goal: $((c \cdot b) \cdot b) \geq (b \cdot (b \cdot c))$

**Theorem 21**
Premises: $(a + d) = a$; $((a + d) + e) \geq 0$; $(b + f) \geq (0 \cdot 0)$
Goal: $((((((c \cdot 0) + (0 \cdot 0)) + (a + d)) \cdot ((0 + ((c + 0) \cdot (a + (-a)))) + a)) \cdot ((a + d) + e)) + (b + f)) \geq ((0 \cdot ((a + d) + e)) + (0 \cdot 0))$

**Theorem 22**
Premises: $(c + d) \geq 0$; $((0 + 0) + e) \geq (0 + 0)$
Goal: $(((((0 + (c + (-c))) \cdot (-c)) \cdot (\frac{1}{((0 \cdot (-c)) + (0 \cdot (-c)))})) \cdot (0 + 1)) \cdot (c + d)) + ((0 + 0) + e)) \geq ((0 \cdot (c + d)) + (0 + 0))$

**Theorem 23**
Premises: $((a^2) + d) \geq 0$
Goal: $((((a \cdot a) + c) \cdot (0 + (1 \cdot (a \cdot a)))) \cdot ((a^2) + d)) \geq ((((a \cdot a) \cdot ((a^2) + 0)) + (c \cdot ((a^2) + 0))) \cdot ((a^2) + d))$

**Theorem 24**
Premises: $(c + d) = c$; $((0 + a) + e) \geq a$

Goal: $((((a + b) \cdot (((a + (-a)) + (a + b)) + (c + d))) \cdot (((((0 + a) + b) + c) \cdot (a + b)) \cdot 1)) + ((0 + a) + e)) \geq (0 + a)$

**Theorem 25**
Goal: $1 \geq ((a \cdot (c + b)) \cdot (\frac{1}{((a \cdot c) + (a \cdot b))}))$

**Theorem 26**
Premises: $(a + d) \geq b$
Goal: $((0 \cdot ((((((a+c)+a) \cdot (a \cdot c)) \cdot (a \cdot c)) + ((a \cdot c) \cdot (a \cdot c))) + (-((((((a+(c+a)) \cdot a) \cdot c) + (a \cdot c)) \cdot (a \cdot c)) + 0)))) + (a + d)) \geq (0 + b)$

**Theorem 27**
Premises: $((c \cdot b) + d) = (b \cdot b)$; $((b \cdot b) + e) \geq a$
Goal: $((((b + b) + (b + b)) \cdot (((c \cdot (b \cdot b)) + b) + b) + (b^2))) + ((b \cdot b) + e)) \geq ((((b + b) \cdot (((c \cdot b) \cdot b) + (b + b)) + ((c \cdot b) + d))) + ((b + b) \cdot (((c \cdot b) \cdot b) + (b + b)) + ((c \cdot b) + d)))) + a)$

**Theorem 28**
Premises: $((b \cdot 0) + d) \geq c$
Goal: $((((b + (((0 + c) + (0 + c)) + 0)) \cdot 0) \cdot ((b \cdot 0) + (((0 + c) + (0 + c)) \cdot 0))) + ((b \cdot 0) + d)) \geq (0 + c)$

**Theorem 29**
Premises: $(a + d) \geq 0$
Goal: $((0 \cdot (((( c \cdot c) + (c \cdot 0)) \cdot a) + (-(((c + 0) \cdot ((c + 0) \cdot a)) \cdot 1)))) + (a + d)) \geq (0 + 0)$

**Theorem 30**
Premises: $(a + d) \geq c$
Goal: $(((b \cdot (b \cdot 1)) + (b \cdot c)) + (a + d)) \geq ((0 + (b \cdot ((b \cdot 1) + c))) + c)$

**Theorem 31**
Goal: $(0 + (0 + (c + b))) \geq (0 + ((b + c) + 0))$

**Theorem 32**
Goal: $(a + (a + 0)) \geq ((((0 + a) + 0) + a) + 0)$

**Theorem 33**
Premises: $((c + c) + d) \geq a$; $(d + e) \geq 0$; $((c + c) + f) \geq (0 + a)$; $(b + g) \geq 0$
Goal: $(((((((c+c)+(c+c)) \cdot ((c+c)+(c+c))) + ((c+c)+d)) + (d+e)) + ((c+c)+f)) + (b+g)) \geq ((((0+a)+0)+(0+a))+0)$

**Theorem 34**
Goal: $(((0 + b) + c) + a) \geq (0 + (0 + (b + (c + a))))$

**Theorem 35**
Premises: $(a + d) \geq 0$; $(a + e) \geq (c \cdot c)$; $(e + f) \geq 0$; $(c + g) \geq 0$; $(c + h) \geq (c + g)$; $(c + i) \geq 0$
Goal: $(((((((c \cdot c) \cdot (a+d)) + (a+e)) \cdot (e+f)) \cdot (c+g)) + (c+h)) \cdot (c+i)) \geq ((((((0 \cdot (a+d)) + (c \cdot c)) \cdot (e+f)) \cdot (c+g)) + (c+g)) \cdot (c+i))$

**Theorem 36**
Goal: $(1 \cdot (1 \cdot (1 \cdot a))) \geq (1 \cdot ((a + 0) + 0))$

**Theorem 37**
Premises: $(b + d) \geq b$; $((c + b) + e) \geq c$; $(b + f) \geq a$; $(e + g) \geq (b + f)$
Goal: $(((c + (b + d)) + (b + f)) + (e + g)) \geq (((((c + b) + c) + (-((c + b) + e))) + a) + (b + f))$

**Theorem 38**
Goal: $((a + (((b+c) \cdot (b+c)) + ((c+b) \cdot b))) \cdot ((c+b) + (c+b))) \geq ((((((b+c) \cdot (c+b)) + ((b+c) \cdot b)) + a) \cdot (c+b)) + (((((b+c) \cdot (c+b)) + ((b+c) \cdot b)) + a) \cdot (c+b)))$

**Theorem 39**
Premises: $(c+d) = b$; $((c+b) + e) = (c+d)$; $(a+f) \geq 0$; $(0+g) \geq 0$; $(g+h) \geq 0$; $(d+i) \geq 0$
Goal: $((((((c+(c+d)) + ((c+b)+e)) \cdot (a+f)) \cdot (0+g)) \cdot (g+h)) \cdot (d+i)) \geq ((((((c+b)+(c+d)) \cdot (a+f)) \cdot (0+g)) \cdot (g+h)) \cdot (d+i))$

**Theorem 40**
Goal: $((((c+a) \cdot b) \cdot b) + (a+c)) \geq ((a+c) + (((a+c) \cdot b) \cdot b))$

**Theorem 41**
Goal: $(((c+b) + (a+(c+b))) \cdot (\frac{1}{((((1 \cdot c)+b)+a)+(c+b))})) \geq (1 \cdot 1)$

**Theorem 42**
Premises: $(c+d) = b$
Goal: $(((((c \cdot b)+(c^2)) \cdot ((b+c) \cdot (c \cdot b))) + (c+d)) \cdot (((((c \cdot (b+c)) \cdot (b+c)) \cdot c) \cdot b) + b)) \geq (((((c \cdot b)+(c^2)) \cdot ((b+c) \cdot (c \cdot b))) + (c+d))^2)$

**Theorem 43**
Premises: $(a+d) = b$; $(d+e) = a$; $(c+f) \geq 0$; $((b+b)+g) \geq 0$
Goal: $((1 \cdot (c+f)) \cdot ((b+b)+g)) \geq (((((b+b)+a) \cdot (\frac{1}{(0+((b+(a+d))+(d+e)))})) \cdot (c+f)) \cdot ((b+b)+g))$

**Theorem 44**
Goal: $(((((a \cdot 1) \cdot a) \cdot 1) \cdot b) + (((a \cdot 1) \cdot (a \cdot 1)) \cdot (a \cdot a))) \geq (1 \cdot ((((a \cdot a) \cdot 1) \cdot b) + (((a \cdot a) \cdot 1) \cdot (a \cdot a))))$

**Theorem 45**
Premises: $((c+0) + d) \geq b$; $(1+e) \geq a$
Goal: $((0 + ((c+0) + d)) + (1+e)) \geq (((0 + (-((c \cdot 1) + (-(c+0))))) + b) + a)$

**Theorem 46**
Premises: $(c+d) \geq (a \cdot c)$
Goal: $(((1 \cdot (1 \cdot (a \cdot (a \cdot c)))) \cdot ((1 \cdot ((a \cdot a) \cdot c)) + 0)) + (c+d)) \geq (0 + (a \cdot c))$

**Theorem 47**
Premises: $(c+d) \geq c$
Goal: $((c \cdot (0+c))^2) \geq (((0 + ((c \cdot (0+c)) \cdot (c^2))) + c) + (-(c+d)))$

**Theorem 48**
Premises: $(a+d) = b$
Goal: $(1 \cdot ((b+b) + (-(1 \cdot (b+(a+d)))))) \geq (1 \cdot (0 \cdot 1))$

**Theorem 49**
Premises: $((c \cdot b) + d) = a$; $((c \cdot b) + e) \geq b$
Goal: $(((b \cdot b) \cdot (a \cdot (c \cdot b))) + ((c \cdot b) + e)) \geq ((((b \cdot b) \cdot a) \cdot (c \cdot b)) + b)$

**Theorem 50**
Goal: $(((a+c) \cdot (c+a)) + ((a \cdot (c+a)) + ((c \cdot c) + (c \cdot a)))) \geq (((a+c) \cdot ((c+a) + (c+a))) \cdot 1)$

