# OpenReview forum: "INT: An Inequality Benchmark for Evaluating Generalization in Theorem Proving"
_ICLR.cc/2021/Conference — ICLR 2021 Poster_

### Official Review · AnonReviewer1 · 2020-10-26
**Synthetic dataset generator for inequality statements over ordered fields**

**Rating:** 6
**Confidence:** 2

**Review:**

The paper describes a synthetic dataset generator for inequality
statements over ordered fields.  The reason is to provide a test of
generalization ability of models in interactive theorem proving tasks
with Lean examples given as motivation. A lightweight syntax tree
based prover is used by the machine learning agents to attempt solving
the generated statements. GNN experiments with various masures show
some generalization ability.

I appreciate the developed inequality theorem generator and the
provide API. They indeed could be useful for further investigations.

I am however still not convinced that the considered measures
correspond well to the generalization ability that the paper wants to
show. For me the task is very simple in comparison with the claims.
I am happy about the one given generalization example, but other than
this what I can read from the paper is agents can solve test problems
similar to those seen during training. As such, I do not think that the
research done in the paper supports the conclusions the authors draw.
The same holds for the out-of-distribution generalization where again
the experiments show a small generalization capability but I am not
convinced that this approach can in general lead to generalization.

I appreciate the addition of the Monte Carlo tree search to the base
model and the consideration of some models (even if still short).

---

> ### Author Response · Authors · 2020-11-18
> **Thank you for your encouragement and we believe INT has great values for investigating generalization in theorem proving.**
>
> We thank the reviewer for their efforts and inputs. We are glad that the reviewer thinks the method proposed can be useful. In the following we address specific questions raised by the reviewer:
>
> “I am however still not convinced that the considered measures correspond well to the generalization ability that the paper wants to show. For me the task is very simple in comparison with the claims … I am not convinced that this approach can in general lead to generalization.”
>
> We believe the conclusions we draw on INT have the potential to generalize to real-world math data, and INT has great value in catalyzing the progress of realistic theorem proving with its simplicity. We argue in the following three perspectives.
>
>  - *INT has been shown to improve theorem provers on real-world proof corpora (MetaMath)*. In [2], the authors adopted our data generation method and augmented their dataset with synthetic proofs. The experiments showed a 2.28% improvement with their 700m model in success rates. This demonstrates that the INT generation method and dataset are already helping formal mathematics.
>
>   - *INT can be used to sanity check our algorithms*: Theorem proving is a novel problem to the field of machine learning, presenting many new challenges, all of which current technology has difficulty to address. These challenges include out-of-distribution generalization, hopeless exploration due to the large action space, long horizon, and jumpy planning, and others. Because INT closely mimics the interface of the modern ITPs, many of these challenges can be investigated in INT. Due to the lower complexity of INT than real-world math problems, we expect one can do a quick sanity check of the algorithm before investing time into more complex problems. We agree that successful methods on INT does not guarantee generalization to real-world math data, but we cannot believe an algorithm to work well on real-world math data before it solves INT. Just like the MNIST vs ImageNet analogy, we cannot expect a model to perform well on ImageNet before it solves MNIST. Notice that novel and influential techniques such as the combination of backpropagation and convolution were first introduced [1] when MNIST had not been solved. We hope INT can serve the same role as MNIST for the theorem proving challenge.
>
>   - *INT provides a lightweight open-sourced infrastructure to accelerate scientific iteration*. Training deep neural networks to do theorem proving can incur large computation and time costs. For instance, two papers [2, 3] described training on two lightweight ITP datasets HOList and MetaMath respectively. In [3], reinforcement learning was used. It takes over 25 CPU years and 40 GPU days to train a single reinforcement learning loop. In [2], ~1000 GPU hours are needed to evaluate the model alone. Moreover, these projects are not open-sourced, making it more difficult to reproduce the experiment and follow up on this research. INT is however a much more lightweight environment. It is easy to set up the environment: everything is written in python (unlike the traditional ITPs are written in Ocaml/SML). It provides 6X faster simulation speed than HOL Light (as mentioned in section 3.2). Furthermore, all of our experimental code will be open-sourced, which allows everyone to reproduce with affordable compute resources. These advantages of INT makes it a great candidate to perform initial experiments to sanity check new algorithms and accelerate scientific iterations.
>
> “but other than this what I can read from the paper is agents can solve test problems similar to those seen during training...”
>
> Besides demonstrating “the agent can solve test problems similar to those seen during training”, we have also other interesting observations from INT. Our findings include: (1) The neural agents generalizes better when the training distribution becomes more diverse. This is demonstrated clearly through generalization across axiom orders and axiom combinations. (2) Neural agents still struggle at generalization along the axiom number and proof length dimensions. (3) Transformer models remarkably achieve much stronger absolute performance than all the other models, however they often obtain a larger out-of-distribution generalization gap than GNNs, especially along the initial condition dimension. (4) Neural agents can benefit from using MCTS in the test time to achieve higher success rates.
>
> [1] Yann LeCun, Bernhard E. Boser, John S. Denker, Donnie Henderson, Richard E. Howard, Wayne E. Hubbard, Lawrence D. Jackel: Backpropagation Applied to Handwritten Zip Code Recognition. Neural Comput. 1(4): 541-551 (1989)
>
> [2] Stanislas Polu, Ilya Sutskever: Generative Language Modeling for Automated Theorem Proving. CoRR abs/2009.03393 (2020)
>
> [3] Kshitij Bansal, Sarah M. Loos, Markus N. Rabe, Christian Szegedy: Learning to Reason in Large Theories without Imitation. CoRR abs/1905.10501 (2019)

---

> > ### Author Response · Authors · 2020-11-23
> > **Does our rebuttal address your concern?**
> >
> > We wonder if our rebuttal have addressed your concerns. If not, we are happy to clarify further. Please let us know.
> > Thank you very much.

---

### Official Review · AnonReviewer4 · 2020-10-28
**A lot of work but in unclear direction**

**Rating:** 6
**Confidence:** 4

**Review:**

The paper introduces a synthetic dataset for learning theorem proving
based on (ordered) field axioms. It also trains learning-based theorem
proving on them and in particular compares transformers and GNNs.  The
motivation is to provide a dataset for investigating the
generalization ability of various ML models.  Six different metrics
are used for the experiments.

The generator and the prover look nice and cleanly implemented and a lot of
experiments have been done.

On the other hand, it is not quite clear how the six metrics actually
reflect generalization ability on theorems encountered in practice,
and what new insights are gained from the experiments.

The dataset also looks quite simple and seems comparable to previous
synthetic arithmetical datasets released e.g. as part of [1]. Other
synthetic datasets were released as part of [2,3] where the tasks
belong to those found interesting and previously used by
well-established ATP researchers.

The method for generating problems looks like quite a straightforward
application of the axioms to an initial statement. I would
think that using actual state-of-the-art ATPs for such forward runs is
probably easier and applicable to an arbitrary domain. Such things have
been done in the past - e.g. to generate the AIM dataset in CASC
2016 [23] by sampling increasingly long inference chains from Prover9
runs on actual mathematically interesting problems.

A great danger of synthetic datasets is that it is completely
unclear how useful the methods are on real-world math data. The
argument that there are not enough real-world math data is flawed. All
of the larger ITP corpora (Isabelle, Mizar, HOl, Coq) are capable of
easily exporting millions of problems and proofs.
Already the first version of MPTP and the AI/TP
experiments based on it [3] allowed and announced a straightforward
generation of 630000 related proof tasks from Mizar. Further millions
of "synthetic" tasks can be created easily by chasing the large
derivation graphs of MML and other ITP libraries. This has been to some
extent used later e.g. in works such as [4,9], where the benefit of
learning from additional proof tasks was also demonstrated.

I would also say that much larger improvements have been achieved by
iterating learning and reasoning and thus "synthesizing" the body of
training proofs.  Such proofs are also created "synthetically" in some
sense - as a result of diversely parameterized proof attempts on a
curriculum of relevant theorems. This goes back at least to MaLARea
[11,12] and continues in systems like ATPBoost [13]. Hence the most
straightforward iterative approach to producing more *real-world*
proof data to learn from seems far from saturated at the moment.

There have also been many ways how to synthesize relevant new
problems/conjectures, some of them neural [15-22].

The comparison of transformers and GNNs is perhaps the most interesting
part of the work. While both settings have been used for ATP tasks
before, this may be their first head-to-head comparison.

Overall, there is a lot of work here, but I am not quite sure how
interesting it is, how much difference it makes to the number of other
methods for generating problems, and if this is going to be a useful dataset.

My feeling is that there has been a lot of "real-world" ATP/ITP datasets out there for a long time, and
that those are very easy to augment further to corpora that will be larger than the number of particles in the universe.
The AI/TP researchers should really focus on developing strong theorem proving methods on them rather than on coming up with more and more datasets, especially very synthetic ones.

I am hesitating between 5 and 6, going now for 5.

Some more detailed comments:
==================

- Formalized mathematical theorems in ITPs include the Feit-
Thompson theorem (Huang et al., 2019)
==> certainly not the right citation?

... the Kepler Conjecture (Bansal et al., 2019).
==> dtto

- Time and memory requirements for the proof assistant have often been an obstacle for using theorem
provers as RL environments. Most existing proof assistants require a large software library to define
numerous mathematical theorems, leading to slow simulation.
==>
What is meant by slow simulation here? Why would a large library lead
to it?  As for RL toolkits, the first RL-for-TP experiments done over
a large part of the Mizar library [5] required orders of magnitude less
resources than e.g. training of AlphaGo/Zero .

- The GNN used does not seem to be logic-aware as the one by Olsak used to extend rlCoP [7] and ENIGMA [8].


[1] Zsolt Zombori, Adrián Csiszárik, Henryk Michalewski, Cezary Kaliszyk, Josef Urban:
Towards Finding Longer Proofs. CoRR abs/1905.13100 (2019)

[2] Thibault Gauthier:
Deep Reinforcement Learning for Synthesizing Functions in Higher-Order Logic. LPAR 2020: 230-248

[3] Thibault Gauthier:
Deep Reinforcement Learning in HOL4. CoRR abs/1910.11797 (2019)

[4] Cezary Kaliszyk, Josef Urban:
Learning-assisted theorem proving with millions of lemmas. J. Symb. Comput. 69: 109-128 (2015)

[5] Cezary Kaliszyk, Josef Urban, Henryk Michalewski, Mirek Olsák:
Reinforcement Learning of Theorem Proving. CoRR abs/1805.07563 (2018)

[6] 	Jan Jakubuv, Josef Urban:
Hammering Mizar by Learning Clause Guidance. CoRR abs/1904.01677 (2019)

[7] Miroslav Olsák, Cezary Kaliszyk, Josef Urban:
Property Invariant Embedding for Automated Reasoning. ECAI 2020: 1395-1402

[8] 	Jan Jakubuv, Karel Chvalovský, Miroslav Olsák, Bartosz Piotrowski, Martin Suda, Josef Urban:
ENIGMA Anonymous: Symbol-Independent Inference Guiding Machine (System Description). IJCAR (2) 2020: 448-463

[9] Kaliszyk, C., Urban, J. & Vyskocil, J. Lemmatization for Stronger Reasoning in Large Theories in
FroCoS 2015 9322 (Springer, 2015), 341–356.

[10] Thibault Gauthier, Cezary Kaliszyk:
Sharing HOL4 and HOL Light Proof Knowledge. LPAR 2015: 372-386

[11] Urban, J. MaLARea: a Metasystem for Automated Reasoning in Large Theories in CADE-21 Work-
shop on Empirically Successful Automated Reasoning in Large Theories 257 (CEUR-WS.org, 2007).

[12] Urban, J., Sutcliffe, G., Pudlák, P. & Vyskocil, J. MaLARea SG1- Machine Learner for Automated
Reasoning with Semantic Guidance in IJCAR 2008 5195 (Springer, 2008), 441–456.

[13] Piotrowski, B. & Urban, J. ATPBoost: Learning Premise Selection in Binary Setting with ATP
Feedback in IJCAR 2018 10900 (Springer, 2018), 566–574.

[14] Gauthier, T., Kaliszyk, C. & Urban, J. TacticToe: Learning to Reason with HOL4 Tactics in LPAR
21 46 (EasyChair, 2017), 125–143.

[15] Gauthier, T., Kaliszyk, C. & Urban, J. Initial Experiments with Statistical Conjecturing over Large
Formal Corpora in (CICM 2016 - Work in Progress Proceedings 1785 (CEUR-WS.org, 2016), 219–
228.

[16] Thibault Gauthier. Deep reinforcement learning in HOL4. CoRR, abs/1910.11797,
2019.

[17] Karel Chvalovsky, Thibault Gauthier & Josef Urban:
First Experiments with Data Driven Conjecturing, AITP'19, 2019.
http://aitp-conference.org/2019/abstract/AITP_2019_paper_27.pdf

[18] Josef Urban, Jan Jakubuv:
First Neural Conjecturing Datasets and Experiments. CICM 2020: 315-323

[19] Lenat, D.B.: AM: an artificial intelligence approach to discovery in mathematics
as heuristic search. Ph.D thesis, Stanford (1976)

[20] Fajtlowicz, S.: On conjectures of Graffiti. Ann. Discrete Math. 72(1–3), 113–118
(1988)

[21] Colton, S.: Automated Theory Formation in Pure Mathematics. Distinguished
Dissertations. Springer, London (2012).

[22] Johansson, M., Rosén, D., Smallbone, N., Claessen, K.: Hipster: integrating theory
exploration in a proof assistant. In: Watt, S.M., Davenport, J.H., Sexton, A.P.,
Sojka, P., Urban, J. (eds.) CICM 2014. LNCS (LNAI), vol. 8543, pp. 108–122.
Springer, Cham (2014).

[23] Geoff Sutcliffe: The 8th IJCAR automated theorem proving system competition - CASC-J8. AI Commun. 29(5): 607-619 (2016)

==================

UPDATE

Thanks to the authors for their detailed replies and paper updates.

I am still quite skeptical about the necessity of the use of INT for the 2% improvement of the Metamath GPT experiments. As I mentioned, just using the millions/billions of internal ITP/ATP lemmas has raised the performance much more in previous experiments. And given the "feed our GPT anything mathy" approach, I would expect that training on any sufficiently big ATP corpus might help the GPT.

In general, I am also still quite skeptical that generating more and more synthetic corpora without good relation to real-world math (and motivation by it) has much meaning and will lead to much progress in the ML-for-TP area.

The fact that some teams have burnt a lot of CPU/GPU power on dubious AI/TP setups does not mean that this is the way how things should be done (and how good AITP research is actually done in many cases). In fact, ATP is quite a good example of a domain where brute force helps only to a very limited extent. Rather than being impressed by the wasted resources, I would advise the authors to focus on resource-controlled setups and competitions such as CASC and CASC LTB.

All this said, I do appreciate the relatively large amount of work the authors have invested in this research. So I increase the score and vote for the paper being published mostly for that reason, while encouraging the authors to focus their energy on more realistic setups.

---

> ### Author Response · Authors · 2020-11-18
> **Thank you for your useful suggestions and we believe INT is valuable to advance research in real-world formal mathematics, due to its simplicity, lightweight-ness, and evidence from GPT-f. Rebuttal Part I.**
>
> We thank the reviewer for their efforts and inputs. In the following we address specific questions raised by the reviewer:
>
> We thank the reviewer for pointing out existing theorem proving datasets that we missed in the paper and the miscitations. We have updated the paper to include some most related works and corrected citations, this includes [5-22].
>
> "It is not quite clear how the six metrics actually reflect generalization ability on theorems encountered in practice, and what new insights are gained from the experiments."
>
> We argue that the five out-of-distribution generalization dimensions INT are present in real-world mathematics.
> - Proof length: Proof length generalization is evident. Proofs are always of various lengths. Past work [23] also acknowledged this generalization dimension.
> - Axiom order, axiom combination, number of axioms: The proof of a theorem use assumed axioms, or previously established theorems/lemmas. The exact sequence of applications of axioms/lemmas defines a proof. We break down the kinds of variations that might occur in such a sequence into several kinds: the order of applications in the sequence, the set of axioms/lemmas, and the number of unique axioms/lemmas. Each of these variations corresponds to one potential variation of the proof, and thereby a different theorem. Each variation is investigated separately in our paper.
> - Initial condition: In theorem proving, one often applies the same axiom/theorem in different contexts. Hence, the prover needs to generalize to situations when the arguments are changed, but the theorem-to-apply stays the same. This motivates the generalization dimension across initial conditions.
>
> "The dataset also looks quite simple and seems comparable to previous synthetic arithmetical datasets released."
>
> There are notable differences between our dataset and the previously introduced synthetic datasets. Our dataset is mainly introduced to investigate 6 types of generalization, and the theorem generator is specifically designed for this purpose. The past synthetic datasets, e.g., [23] only studies one generalization dimension -- the length of the proof. Furthermore, besides the theorem generator, we also introduced a proof assistant whose interface closely mimics that of modern ITPs. This allows one to conduct sanity checking experiments for algorithms to be used in a realistic ITP environment. This feature was not introduced in any past synthetic dataset.

---

> > ### Author Response · Authors · 2020-11-18
> > **Thank you for your useful suggestions and we believe INT is valuable to advance research in real-world formal mathematics, due to its simplicity, lightweight-ness, and evidence from GPT-f. Rebuttal Part II.**
> >
> > “A great danger of synthetic datasets is that it is completely unclear how useful the methods are on real-world math data.”
> >
> > We would like to INT can advance progress in real-world formal mathematics from 3 perspectives:
> >   - *INT has been shown to improve theorem provers on real-world proof corpora (MetaMath)*. In [2], the authors adopted our data generation method and augmented their dataset with synthetic proofs. The experiments showed a 2.28% improvement with their 700m model in success rates. This demonstrates that the INT generation method and dataset are already helping formal mathematics.
> >
> >   - *INT can be used to sanity check our algorithms*: Theorem proving is a novel problem to the field of machine learning, presenting many new challenges, all of which current technology has difficulty to address. These challenges include out-of-distribution generalization, hopeless exploration due to the large action space, long horizon, and jumpy planning, and others. Because INT closely mimics the interface of the modern ITPs, many of these challenges can be investigated in INT. Due to the lower complexity of INT than real-world math problems, we expect one can do a quick sanity check of the algorithm before investing time into more complex problems. We agree that successful methods on INT does not guarantee generalization to real-world math data, but we cannot believe an algorithm to work well on real-world math data before it solves INT. Just like the MNIST vs ImageNet analogy, we cannot expect a model to perform well on ImageNet before it solves MNIST. Notice that novel and influential techniques such as the combination of backpropagation and convolution were first introduced [1] when MNIST had not been solved. We hope INT can serve the same role as MNIST for the theorem proving challenge.
> >
> >   - *INT provides a lightweight open-sourced infrastructure to accelerate scientific iteration*. Training deep neural networks to do theorem proving can incur large computation and time costs. For instance, two papers [2, 3] described training on two lightweight ITP datasets HOList and MetaMath respectively. In [3], reinforcement learning was used. It takes over 25 CPU years and 40 GPU days to train a single reinforcement learning loop. In [2], ~1000 GPU hours are needed to evaluate the model alone. Moreover, these projects are not open-sourced, making it more difficult to reproduce the experiment and follow up on this research. INT is however a much more lightweight environment. It is easy to set up the environment: everything is written in python (unlike the traditional ITPs are written in Ocaml/SML). It provides 6X faster simulation speed than HOL Light (as mentioned in section 3.2). Furthermore, all of our experimental code will be open-sourced, which allows everyone to reproduce with affordable compute resources. These advantages of INT makes it a great candidate to perform initial experiments to sanity check new algorithms and accelerate scientific iterations.
> >
> > “The argument that there is not enough real-world math data is flawed.”
> >
> > We want to clarify with the reviewer that the lack of real-world math data argument is used to emphasize the importance of studying out-of-distribution generalization. In fact, one can never claim we have enough data for the theorem proving challenge, as interesting theorems always are non-trivially different from those we have proofs for. There is a combinatorial explosion of the number of theorems given a fixed set of axioms, and in no way can math data be enough to cover the entire distribution. This is why we think studying out-of-distribution generalization is important for theorem proving.

---

> > > ### Author Response · Authors · 2020-11-18
> > > **Thank you for your useful suggestions and we believe INT is valuable to advance research in real-world formal mathematics, due to its simplicity, lightweight-ness, and evidence from GPT-f. Rebuttal Part III.**
> > >
> > > “Past work to generate synthetic datasets”
> > >
> > > We thank the reviewer for mentioning using actual state-of-the-art ATPs for generating synthetic runs. However, we have not found in [4], the detailed setup of how synthetic theorems can be generated by sampling increasingly long inference chains from the AIM dataset. We do not find descriptions that allow one to generate diverse theorems as specified by all the generalization dimensions as we study in this paper. We would be grateful if the reviewer could direct us to the work that elaborates this.
> > >
> > > “GNNs used are not logic-aware”
> > >
> > > We chose the vanilla implementation of baselines to fairly compare across neural architectures. Logic-aware graph neural nets could be interesting future work to try on this dataset.
> > >
> > > “Transformer vs. GNNs”:
> > >
> > > We are glad that the reviewer finds the head-to-head comparison between transformers and GNNs interesting.
> > >
> > > We agree with the reviewer that there have been a lot of real-world ATP/ITP datasets and want to reiterate that the contribution of this paper is an infinite data generation method to study out-of-distribution generalization, a lightweight theorem proving environment, and the experimentation and conclusions with current neural architectures. We believe that INT is not “just another dataset”, but a tool to help devise better theorem provers. Our view on what AI/TP researchers should focus on aligns with that of the reviewer.
> > >
> > > [1] Yann LeCun, Bernhard E. Boser, John S. Denker, Donnie Henderson, Richard E. Howard, Wayne E. Hubbard, Lawrence D. Jackel: Backpropagation Applied to Handwritten Zip Code Recognition. Neural Comput. 1(4): 541-551 (1989)
> > >
> > > [2] Stanislas Polu, Ilya Sutskever: Generative Language Modeling for Automated Theorem Proving. CoRR abs/2009.03393 (2020)
> > >
> > > [3] Kshitij Bansal, Sarah M. Loos, Markus N. Rabe, Christian Szegedy: Learning to Reason in Large Theories without Imitation. CoRR abs/1905.10501 (2019)
> > >
> > > [4] Geoff Sutcliffe: The 8th IJCAR automated theorem proving system competition - CASC-J8. AI Commun. 29(5): 607-619 (2016)
> > >
> > > [5] Josef Urban, Jan Jakubuv: First Neural Conjecturing Datasets and Experiments. CICM 2020: 315-323
> > >
> > > [6] Josef Urban: MPTP 0.2: Design, Implementation, and Initial Experiments. J. Autom. Reason. 37(1-2): 21-43 (2006)
> > >
> > > [7] Piotr Rudnicki: An overview of the Mizar project. In Proceedings of the 1992 Workshop on Types for Proofs and Programs, pp. 311-330 (1992)
> > >
> > > [8] Jan Jakubuv, Josef Urban: Hammering Mizar by Learning Clause Guidance. CoRR abs/1904.01677 (2019)
> > >
> > > [9] Miroslav Olsák, Cezary Kaliszyk, Josef Urban: Property Invariant Embedding for Automated Reasoning. ECAI 2020: 1395-1402
> > >
> > > [10] Jan Jakubuv, Karel Chvalovský, Miroslav Olsák, Bartosz Piotrowski, Martin Suda, Josef Urban: ENIGMA Anonymous: Symbol-Independent Inference Guiding Machine (System Description). IJCAR (2) 2020: 448-463
> > >
> > > [11] Cezary Kaliszyk, Josef Urban, Jirí Vyskocil: Lemmatization for Stronger Reasoning in Large Theories. FroCos 2015: 341-356
> > >
> > > [12] Thibault Gauthier, Cezary Kaliszyk: Sharing HOL4 and HOL Light Proof Knowledge. LPAR 2015: 372-386
> > >
> > > [13] Josef Urban: MaLARea: a Metasystem for Automated Reasoning in Large Theories. ESARLT 2007
> > >
> > > [14] Josef Urban, Geoff Sutcliffe, Petr Pudlák, Jirí Vyskocil: MaLARea SG1- Machine Learner for Automated Reasoning with Semantic Guidance. IJCAR 2008: 441-456
> > >
> > > [15] Piotrowski, B. & Urban, J. ATPBoost: Learning Premise Selection in Binary Setting with ATP Feedback in IJCAR 2018 10900 (Springer, 2018), 566–574.
> > >
> > > [16] Gauthier, T., Kaliszyk, C. & Urban, J. TacticToe: Learning to Reason with HOL4 Tactics in LPAR 21 46 (EasyChair, 2017), 125–143.
> > >
> > > [17] Gauthier, T., Kaliszyk, C. & Urban, J. Initial Experiments with Statistical Conjecturing over Large Formal Corpora in (CICM 2016 - Work in Progress Proceedings 1785 (CEUR-WS.org, 2016), 219– 228.
> > >
> > > [18] Karel Chvalovsky, Thibault Gauthier & Josef Urban: First Experiments with Data Driven Conjecturing, AITP'19, 2019. http://aitp-conference.org/2019/abstract/AITP_2019_paper_27.pdf
> > >
> > > [19] Lenat, D.B.: AM: an artificial intelligence approach to discovery in mathematics as heuristic search. Ph.D thesis, Stanford (1976)
> > >
> > > [20] Fajtlowicz, S.: On conjectures of Graffiti. Ann. Discrete Math. 72(1–3), 113–118 (1988)
> > >
> > > [21] Colton, S.: Automated Theory Formation in Pure Mathematics. Distinguished Dissertations. Springer, London (2012).
> > >
> > > [22] Johansson, M., Rosén, D., Smallbone, N., Claessen, K.: Hipster: integrating theory exploration in a proof assistant. In: Watt, S.M., Davenport, J.H., Sexton, A.P., Sojka, P., Urban, J. (eds.) CICM 2014. LNCS (LNAI), vol. 8543, pp. 108–122. Springer, Cham (2014).
> > >
> > > [23] Zsolt Zombori, Adrián Csiszárik, Henryk Michalewski, Cezary Kaliszyk, Josef Urban: Towards Finding Longer Proofs. CoRR abs/1905.13100 (2019)

---

> > > > ### Author Response · Authors · 2020-11-23
> > > > **Do you have any further questions? Thanks!**
> > > >
> > > > We have addressed your questions in our rebuttal. We wonder if they have addressed most of your concerns. If not, we are happy to clarify further.
> > > >
> > > > Many thanks again for your efforts and suggestions. They really have helped us improve the paper.

---

### Official Review · AnonReviewer2 · 2020-10-28
**An approach for evaluating theorem proving and a detailed empirical study of different neural network architectures for theorem proving. The paper addresses an important problem and is well presented.**

**Rating:** 7
**Confidence:** 2

**Review:**

An approach to evaluate theorem proving using neural networks.

The paper describes a systematic approach to evaluating theorem provers in terms of generalization. Specifically, since for theorem proving the test data can be significantly different from the training data, the proposed approach develops a method to generate synthetic problems to simulate this. The main contribution is a theorem generator that can produce non-trivial theorems to evaluate a model.

The paper is well written and has extensive empirical analysis for neural network based theorem proving. It produces a general approach that can be used to improve the state-of-the-art in evaluation of neural network based theorem provers. I am not very familiar with  other approaches that can generate synthetic data for theorem proving, but it seems like this is first approach that can generate an infinite number of theorems with complex proofs which can evaluate generalization of the prover which to me seems to be a significant contribution. Further, the empirical analysis is extensive that includes several well-known methods along several dimensions. I think this paper has sufficient merit with potential for significant follow up research based on the proposed benchmark.

---

> ### Author Response · Authors · 2020-11-18
> **Thank you for your efforts, inputs, and encouraging words.**
>
> We thank the reviewer for your efforts, inputs, and encouraging words. We are glad that the reviewer finds the paper interesting. To the best of our knowledge, our theorem proving dataset is the first theorem proving dataset that is specifically designed to evaluate the generalization abilities of machine learning. We believe the extensive experiments done in this work revealed some weaknesses of current machine learning methods for theorem proving. We also believe that the work holds great potential for significant follow-up research.

---

### Official Review · AnonReviewer3 · 2020-10-31
**Interesting approach to systematically study generalization in theorem proving but some questions about the experiments**

**Rating:** 8
**Confidence:** 4

**Review:**

This paper presents a simple theorem proving framework (supporting a limited logic e.g., no disjunctions) along with a sample set of axioms and algorithms that can be used to generate unlimited training data for proving theorems in this system.

Although the proof system itself is not of much practical interest, I really like this approach since it allows for a much more rigorous way to study the natural generalization questions that arise in theorem proving since the data generation can be controlled to a great extent. (The usual work in this area that uses existing proof corpora does not easily allow for answering such questions due to the lack of such control.)

The authors use this framework to study the generalization of two classes of networks commonly used in theorem proving (transformers and graph neural networks), as well as to explore the utility of MCTS in theorem proving. Furthermore, the ability to generate a large amount of synthetic data is of independent interest, and the authors refer to a recent work that has used their framework to generate training data for a different system.

My main concern about this direction of research would be that perhaps these proofs are just too simple (e.g. like MNIST), and the results on this may not generalize to “real” proof corpora (e.g. ImageNet). But that remains to be seen, and I don’t necessarily consider that a drawback at the present time. However, the authors could bolster their case by doing a survey of other papers in this area which are using real proof databases and see to what extent their results in this simplified setting agree with or disagree with the results there.

Some important issues related to the writeup. Would like these to be addressed by the authors since it may reflect errors in understanding on my part.

1. I found Figure 1 confusing. In (a) (LEAN), how is the matching done i.e. do the rw rules match the LHS or the RHS of the equality? Would be good to explicitly bracketize the a + b + c rather than relying on implied left to right or right to left associativity of + (as is done in (b)).

2. In Figure 1 (c) shouldn’t the red circle for “Addition Associativity” be on the left child of the EQUALS node to provide the same example as (b)? (If so, this should also cause Goal 2 graph to change.)

3. In the worked example, I would encourage the authors to explicitly bracketize the expressions in the RHS of the different steps (to avoid any confusion with associativity).

4. At the bottom of page 3, you say that you initiate the core logic statement $C_0$ to be one of the initial conditions. But in the worked example, the initial conditions are $a = a, b = b,$ etc. So how do you get the expression a + (b + c) in Step 1? Shouldn’t C_0 be a simple equality like $a = a$ or $b = b$ rather than $a + b = a + b$?

Some questions about the experiments:

1. In Section 4.1, the training appears non-standard since you generate 1000 problems at a time and take 10 epochs; then generate another 1000 problems, and so on. Why not generate the 1.5 million problems up front and then train on the entire training set? What happens to the results if you do that?

2. How long does an experiment as set up in Section 4.1 take to run?

3. Related: Why not run for 5 seeds if computationally feasible. What is the variation in your results if you do that (rather than take the best)? I worry that with your current presentation of results, it is not clear what the inherent noise levels are, i.e. are the differences you find (e.g. that GNNs do better than Transformers on out of distribution) statistically significant or not? If 1.5MM examples is too computationally expensive, even running some of your experiments on a smaller set of examples would provide a sense of this variation.

*Post Rebuttal*

Thank you for your responses. I am bumping up the score.

---

> ### Author Response · Authors · 2020-11-18
> **Thank you for your useful suggestions and we believe there are great values in INT with its simplicity. Rebuttal Part I.**
>
> We thank the reviewer for your efforts and inputs. We are glad that the reviewer finds the paper interesting. The following are our answers to specific questions:
>
> “My main concern about this direction of research would be that perhaps these proofs are just too simple (e.g. like MNIST), and the results on this may not generalize to “real” proof corpora (e.g. ImageNet)...”:
>
> We would like to argue that there are great values in INT for catalyzing research with its simplicity from 3 perspectives:
>   - *INT has been shown to improve theorem provers on real-world proof corpora (MetaMath)*. In [2], the authors adopted our data generation method and augmented their dataset with synthetic proofs. The experiments showed a 2.28% improvement with their 700m model in success rates. This demonstrates that the INT generation method and dataset are already helping formal mathematics.
>
>   - *INT can be used to sanity check our algorithms*: Theorem proving is a novel problem to the field of machine learning, presenting many new challenges, all of which current technology has difficulty to address. These challenges include out-of-distribution generalization, hopeless exploration due to the large action space, long horizon, and jumpy planning, and others. Because INT closely mimics the interface of the modern ITPs, many of these challenges can be investigated in INT. Due to the lower complexity of INT than real-world math problems, we expect one can do a quick sanity check of the algorithm before investing time into more complex problems. We agree that successful methods on INT does not guarantee generalization to real-world math data, but we cannot believe an algorithm to work well on real-world math data before it solves INT. Just like the MNIST vs ImageNet analogy, we cannot expect a model to perform well on ImageNet before it solves MNIST. Notice that novel and influential techniques such as the combination of backpropagation and convolution were first introduced [1] when MNIST had not been solved. We hope INT can serve the same role as MNIST for the theorem proving challenge.
>
>   - *INT provides a lightweight open-sourced infrastructure to accelerate scientific iteration*. Training deep neural networks to do theorem proving can incur large computation and time costs. For instance, two papers [2, 3] described training on two lightweight ITP datasets HOList and MetaMath respectively. In [3], reinforcement learning was used. It takes over 25 CPU years and 40 GPU days to train a single reinforcement learning loop. In [2], ~1000 GPU hours are needed to evaluate the model alone. Moreover, these projects are not open-sourced, making it more difficult to reproduce the experiment and follow up on this research. INT is however a much more lightweight environment. It is easy to set up the environment: everything is written in python (unlike the traditional ITPs are written in Ocaml/SML). It provides 6X faster simulation speed than HOL Light (as mentioned in section 3.2). Furthermore, all of our experimental code will be open-sourced, which allows everyone to reproduce with affordable compute resources. These advantages of INT makes it a great candidate to perform initial experiments to sanity check new algorithms and accelerate scientific iterations.
>
> “However, the authors could bolster their case by doing a survey of other papers in this area which are using real proof databases and see to what extent their results in this simplified setting agree with or disagree with the results there. ”
>
> We appreciate the suggestion to conduct a survey on real proof corpora and compare conclusions. We surveyed prior works for consistency in two types of results: architecture comparisons and systematic generalization.
>   - Architecture: In [4], the authors showed that WaveNets have superior performances than TreeRNNs and TreeLSTMs, but are inferior to GNNs in [5,6]. These conclusions are consistent with our observation in the architecture comparison that GNNs are slightly better than TreeLSTMs. We could not find any past work comparing GNNs versus Transformers. Our work is the first time when transformers and GNNs are compared on the same theorem proving the dataset.
>   - Systematic generalization: We could not find any past results on systematic generalization on realistic human datasets. We believe this is due to the scarcity of formalized human corpus, and hence there are no sufficient data points to do well-controlled experiments to benchmark these systematic generalization dimensions. The only past work on systematic generalization for theorem proving is [7], in which generalization to longer proofs (proof length dimension) was discussed. But unfortunately, their experiments were also conducted in synthetic datasets.

---

> > ### Author Response · Authors · 2020-11-18
> > **Thank you for your useful suggestions and we believe there are great values in INT with its simplicity. Rebuttal Part II.**
> >
> > We would like to thank the reviewer for providing many writing suggestions. Here are some clarifications. We also incorporated your suggestions in our newest version.
> >  - In the LEAN theorem prover, the rewrite tactic matching is done in a first-matched-first-serve fashion. In this case the first line “rw add_comm (a+b);” captures the whole LHS in the objective and rewrites it to (c+(a+b)). Thank you for pointing out this source of confusion. We will properly bracketize the equations to eliminate any ambiguity.
> >  - There is indeed an error in figure 1. It is actually in figure 1(b), where a correct but different proof was expressed. The rewrite tactic with the add_assoc argument in LEAN as well as the Addition Associativity both express the rewriting of ((x+y)+z) to (x+(y+z)). We will correct figure 1(b) and update figure 1(a) to include the goal after each step so that confusion can be avoided.
> >  - We will bracketize the expressions in the worked example where it might cause confusion.
> >  - There is indeed a mistake in the worked example. The correct $C_0$ should be uniformly sampled from the initial conditions {a=a, b=b, …} and in this case $C_0$ is a=a. You can refer to Appendix D for the extension function of Addition Associativity. In the first step of the worked example, we sample two nodes ($n_1$ and $n_2$) from the existing graphs: b and c. Then we synthesize the logic statement $RHS(C_0) + (n_1 + n_2) = (LHS(C_0) + n_1) + n_2: a + (b + c) = (a + b) + c$.
> >
> >
> > We also provide some clarifications for your questions on experiments.
> >  - As the paper mainly focused on systematic/out-of-distribution generalization, we would like to avoid any in-distribution overfitting due to the lack of training data. Hence, we deliberately set up the experiments by generating data points on-the-fly, equivalent to a theoretically infinite dataset. This is also a realistic setting in theorem proving, as the agent can iteratively learn and reason to obtain an infinite amount of experience from exploration in a theorem proving environment. We will generate all the data upfront, run experiments, and include the results in the next version of our paper. We expect very similar results if we run experiments this way. We have not tried to use a smaller dataset, and potentially that will have problems of in-distribution overfitting.
> >  - If running in an online fashion, a single experiment takes about 50 hours to finish on a P100 GPU. If the data generation time is excluded, the experiments take about 25 hours. If running on T4 GPUs, they take slightly shorter (about 20 hours).
> >  - There are 132 experiments in the paper (including those whose results are in Appendix G) if we use one single random seed. Running 5 random seeds on all of them will be taking ~16.5K GPU hours and not feasible for our computational resources. However, we agree that it would be good to verify the conclusions reached, even if just on a smaller scale. Therefore we have run experiments in the proof length generalization (last part of section 4.3) with 5 random seeds and we found that the standard deviations of agents trained with different seeds are between 0.4%-1.7%, while conclusions drawn in this part are based on success rate differences larger than 10% between two agents. We consider the conclusions in this part statistically significant. We update the paper with the newly obtained results and their standard deviations. We will include a paragraph detailing variation in experiments in the camera-ready version.
> >
> > [1] Yann LeCun, Bernhard E. Boser, John S. Denker, Donnie Henderson, Richard E. Howard, Wayne E. Hubbard, Lawrence D. Jackel: Backpropagation Applied to Handwritten Zip Code Recognition. Neural Comput. 1(4): 541-551 (1989)
> >
> > [2] Stanislas Polu, Ilya Sutskever: Generative Language Modeling for Automated Theorem Proving. CoRR abs/2009.03393 (2020)
> >
> > [3] Kshitij Bansal, Sarah M. Loos, Markus N. Rabe, Christian Szegedy: Learning to Reason in Large Theories without Imitation. CoRR abs/1905.10501 (2019)
> >
> > [4] Sarah M. Loos, Geoffrey Irving, Christian Szegedy, Cezary Kaliszyk:
> > Deep Network Guided Proof Search. LPAR 2017: 85-105
> >
> > [5] Kshitij Bansal, Sarah M. Loos, Markus N. Rabe, Christian Szegedy, Stewart Wilcox:
> > HOList: An Environment for Machine Learning of Higher Order Logic Theorem Proving. ICML 2019: 454-463
> >
> > [6] Aditya Paliwal, Sarah M. Loos, Markus N. Rabe, Kshitij Bansal, Christian Szegedy:
> > Graph Representations for Higher-Order Logic and Theorem Proving. AAAI 2020: 2967-2974
> >
> > [7] Zsolt Zombori, Adrián Csiszárik, Henryk Michalewski, Cezary Kaliszyk, Josef Urban: Towards Finding Longer Proofs. CoRR abs/1905.13100 (2019)

---

### Author Response · Authors · 2020-11-18
**Thank all reviewer for your reviews, suggestions, and encouragement. We have updated the paper according to your suggestions.**

We have updated the paper according to reviewers' suggestions. Here we provide a paper update list as follows:
- *Section 2 Related Works*: We corrected the quotations for the formalization of the Feit-Thompson theorem and the Kepler conjecture. We have also updated the section to include more prior works on formal mathematical libraries.
- *Subsection Appendix G.2*: Performance variation on trained agents) We tabled the experimental results of proof length generalization from subsection 4.3, run with 5 random seeds, together with the standard deviations.
- *Subsection 3.2 INT assistant*: We bracketized the proof in the LEAN theorem prover to avoid confusion and corrected the typo in subfigure (b).
- *Subsection 3.3 Theorem generator*: We bracketized terms and corrected the initial conditions’ mistake in the worked example.

---

### Decision · Program_Chairs · 2021-01-07
**Final Decision**

**Decision:**

Accept (Poster)

**Comment:**

This work proposes an algorithm for generating training data to train automatic theorem proving models. In particular, it allows users to pose specific generalization challenges to theorem proving models and evaluate their performance. In doing so, it provides a degree of control over the task space that is greater than when working with 'organic' corpora of real theorems and proofs.

 The authors demonstrated the utility of their generated data by training well-known models such as transformers and GNNs, and were able to derive insights such as the value of MCTS-style planning for finding proofs in particular settings.

After the rebuttal period, all authors agreed that the work was well executed and that the algorithm creates datasets that will be of value to the (learning-based) theorem proving community. As such, they all recommended acceptance to a greater or lesser degree. I am convinced by their arguments, because I think there is real value in using controlled synthetic data alongside real data when making scientific progress on hard problems like theorem proving. I am particularly convinced by the observation that the data generated by this method has already led to improved performance on a real corpus of proofs, as the authors state in their rebuttal. If they have not done so already, I encourage the authors to report this fact in the camera ready version of their paper citing the relevant work.